# H+- and Na+- elicited rapid changes of the microtubule cytoskeleton in the biflagellated green alga *Chlamydomonas*

Yi Liu[1], Mike Visetsouk[1], Michelle Mynlieff[1], Hongmin Qin[2], Karl F Lechtreck[3], Pinfen Yang[1]*

[1]Department of Biological Sciences, Marquette University, Milwaukee, United States; [2]Department of Biology, Texas A&M University, College Station, United States; [3]Department of Cellular Biology, University of Georgia, Athen, United States

**Abstract** Although microtubules are known for dynamic instability, the dynamicity is considered to be tightly controlled to support a variety of cellular processes. Yet diverse evidence suggests that this is not applicable to *Chlamydomonas*, a biflagellate fresh water green alga, but intense autofluorescence from photosynthesis pigments has hindered the investigation. By expressing a bright fluorescent reporter protein at the endogenous level, we demonstrate in real time discreet sweeping changes in algal microtubules elicited by rises of intracellular H+ and Na+. These results from this model organism with characteristics of animal and plant cells provide novel explanations regarding how pH may drive cellular processes; how plants may respond to, and perhaps sense stresses; and how organisms with a similar sensitive cytoskeleton may be susceptible to environmental changes.

DOI: https://doi.org/10.7554/eLife.26002.001

## Introduction

The microtubule (MT) cytoskeletal system is integral to many crucial processes in eukaryotic cells. The opposing ends of these cylindrical polymers exhibit distinct properties, with the plus end growing and shrinking stochastically. MTs establish the polarity of cells and serve as tracks for positioning and trafficking intracellular components. They could also form complex machineries. One example is the mitotic apparatus enabling accurate segregation of chromosomes during mitosis and meiosis. These actions require harnessing MT dynamic instability and the involvement of a wide array of accessory proteins and various post-translational modifications (reviewed by *Hashimoto, 2015*; *Song and Brady, 2015*). Not surprisingly, MTs are the target of natural toxins, such as Taxol (*Weaver, 2014*), and a number of herbicides (reviewed by *Hashimoto, 2015*). Notably, biotic and abiotic stresses alter plant MTs, while Taxol exacerbates stress-induced maldevelopment of seedlings (reviewed by *Wang et al., 2011a*; *Hardham, 2013*; *Oda, 2015*; *Hepler, 2016*). Yet the mechanisms underlying stress-induced changes of MTs and the broad implications remain elusive.

*Chlamydomonas reinhardtii* is uniquely suited for addressing these issues. The unicellular fresh water green alga has signature features of both animal and plant cells (*Merchant et al., 2007*). Like animal cells, it has motile flagella that contain stable MT bundles. Like typical plant cells, the alga is equipped with vacuoles, chloroplast, and dynamic cortical MTs that serve as tracks for the delivery of the enzymes which synthesize the cell wall (*Paredez et al., 2006*). Curiously, its MT system appears susceptible to environmental changes. Its flagella sever readily when the aqueous environment changes suddenly (*Lefebvre et al., 1978*); reviewed by *Quarmby, 2009*). The best characterized stimulus is pH shock. Organic acid, such as acetic acid (HA) diffuses across the plasma membrane and then becomes ionized. The resulting acidification in the cytosol triggers nearly

*For correspondence: pinfen.yang@marquette.edu

Competing interests: The authors declare that no competing interests exist.

simultaneous influx of extracellular $Ca^{2+}$, which signals flagella amputation (*Quarmby, 1996*; *Wheeler et al., 2008*; *Hilton et al., 2016*). Cortical MTs become fewer and shorter, detected by immunofluorescence and biochemistry 5 min after pH shock (*Wang et al., 2013*).

While live cell imaging will be ideal for revealing these acid-induced responses with higher spatial and temporal resolution, autofluorescence from photosynthetic pigments in chloroplast obscures commonly used fluorescent reporters (*Lang et al., 1991*; *Rasala et al., 2013*). Recently, we succeeded in revealing dynamic cortical MTs by taking advantage of the new fluorescent protein, Neon-Green (NG) that is 2.7 X brighter than EGFP (*Shaner et al., 2013*), and the relative abundant plus end-binding protein, EB1, as the NG carrier (*Harris et al., 2016*).

EB1 plays central roles in eukaryotes (*Su et al., 1995*; reviewed by *Akhmanova and Steinmetz, 2010*; *Kumar and Wittmann, 2012*). Its N-terminal domain preferentially binds to the lattice among tubulins at the plus end of MTs, whereas its C-terminal domain can associate with a wide array of proteins. The two domains operate in concert to accelerate MT dynamics (*Rogers et al., 2002*; *Vitre et al., 2008*; *Maurer et al., 2014*) and recruit various+ TIP effector proteins that function at the plus end. In live cell imaging, fluorescent EB1 exhibits a comet pattern seemingly leading the plus end of nascent growing MTs, where tubulins transition from the GTP state to the GDP state (*Maurer et al., 2012*; *Zanic et al., 2009*; reviewed by *Gardner et al., 2013*). As such fluorescent EB1 has been commonly used to report newly generated growing MTs (*Piehl et al., 2004*; *Matov et al., 2010*). However, binding to the GDP zone increases in a number of scenarios (e.g. *Tirnauer et al., 2002*; *Goldspink et al., 2013*; *Tortosa et al., 2013*; *Sayas and Ávila, 2014*). What causes the switch remains uncertain.

Using EB1-NG as a reporter, we captured in real time unexpected changes in EB1-NG patterns and MT dynamics signaled through $H^+$ and $Na^+$. The remarkable sensitivity and the distinct responses in wild type (WT) cells and mutants shed critical insight on the divergence of the MT system, pH regulated processes and the vulnerability of organisms subjected to environmental stresses.

## Results

### EB1-NG reports remarkable sensitivity of the MT system in *Chlamydomonas*

Flagellated *Chlamydomonas* cells in interphase contain two mature basal bodies (BBs) and two pro-basal bodies (*Figure 1a*, top panel) that are templated on mature BBs. Each of mature BBs that are derived from centrioles following cell division nucleates the assembly of the axoneme, a MT-based scaffold that drives the rhythmic beating of the flagellum (*Dutcher and O'Toole, 2016*). Four rootlet MT bundles (thick lines) arrange in a cruciate pattern positioning BBs at the apical end and the other organelles (*Mittelmeier et al., 2011*; *Picariello et al., 2014*). These MT bundles consisting of more than two acetylated stable MTs. In contrast, cortical MTs (thin lines) are singular (*Horst et al., 1999*) and highly dynamic (*Harris et al., 2016*). Under widefield fluorescence microscopy EB1-NG expressed at the level of endogenous EB1 from a genomic construct does not reveal stable MTs except the flagellar tip where plus ends of axonemal MTs undergo turnover continuously (*Harris et al., 2016*). In addition, plus ends of growing cortical MTs appear like the typical comets observed in other eukaryotic cells. Comets emerged from four spots underneath flagella, corresponding to BBs (bottom panel) (*Pedersen et al., 2003*). As shown from the top, side and rear views of cells, comets of nascent cortical MTs travel along the contour of the cell body toward the posterior pole (*Figure 1b*, white arrowhead; *Video 1*). Near the pole comets vanish presumably when MTs stop growing or switch to the shrinking phase. The pattern appears similar in cells resuspended in the commonly used Tris Acetate Phosphate (TAP) culture medium or 10 mM HEPES buffer (containing 5 mM $Na^+$).

The birth of new comets from BBs appeared stochastic. We did not measure the birth rates, hindered by substantial fluctuations and the narrow apical area. Instead we analyzed comet length and speed from the side view. Line scans along the lengths of comets show the typical feature of EB1 comets - the brightest spot corresponds to the area where tubulins are primarily at the transitional state, slightly behind the leading edge of plus ends with GTP-tubulins (*Figure 1c*). The distribution of comet speeds shows that MT growth rates varies nearly two folds (*Figure 1d*). The dataset from cells in the TAP medium (black bars) skews toward the slow end relative to the $Na^+$/HEPES dataset

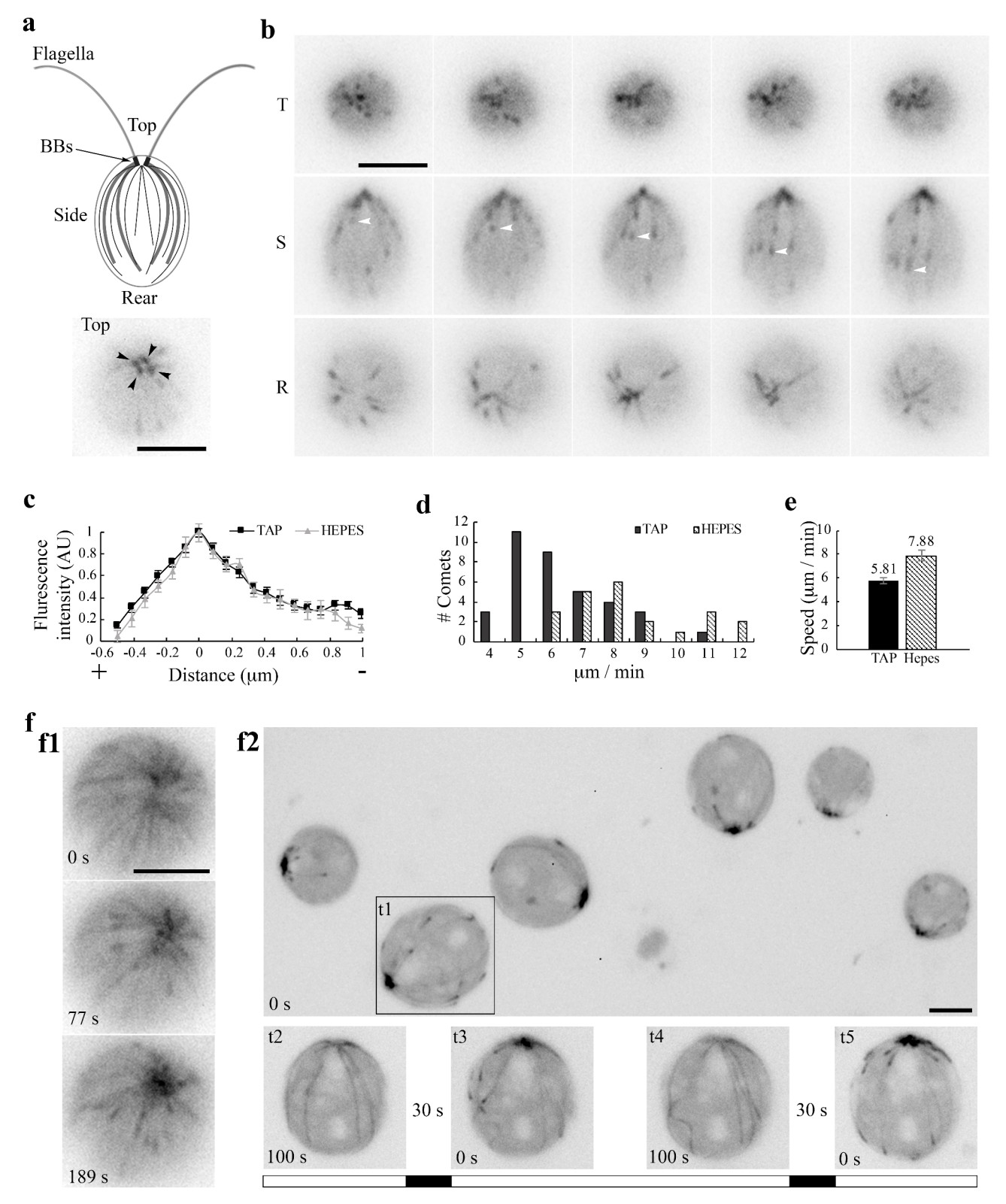

**Figure 1.** EB1 in *Chlamydomonas*. (**a**) A schematic picture depicting flagella and the MT network in the cell body (top panel). Black dots, basal bodies (BB). Thick lines, four stable rootlet microtubule bundles. Thin lines in the cell body, the dynamic cortical MTs. A top view of EB1-NG transgenic cells reveals a pattern that resembles 4 BBs (bottom panel). (**b**) Time-lapse fluorescent images taken 10 s apart from the top (T), side (S) and rear (R) of cells resuspended in the TAP culture medium. EB1-NG appeared like typical comets (arrowheads), emerging from the BB area, coursing along the contour

*Figure 1 continued on next page*

*Figure 1 continued*

of the cell body and then vanishing as approaching the rear end. The frame rate was 1 frame/s. (c) Normalized line scans along the length of MT plus ends showed a similar EB1 intensity profile in the TAP medium (n = 18 comets from 6 cells) and the Na$^+$/HEPES buffer (n = 11 comets from 3 cells). The position with peak intensity was designated as 0. The value was negative toward plus end; positive toward BBs. AU, arbitrary unit of fluorescence intensity. (d) The distribution and (e) the mean and the SEM of EB1 comet speed in the TAP medium (n = 36 comets from 6 cells in 6 recordings) and 5 mM Na$^+$/HEPES buffer (n = 22 comets from 3 cells in 3 recordings) are significant different (Mann-Whitney U test, p<0.001). (f) Altered MT patterns during fluorescence microscopy. The EB1 comet pattern occasionally switched to a bird cage pattern (f1). Comets returned while the bird cage pattern receded in ~ 1 min. In flattened cells that were compressed by the cover slip gradually, both MTs and comets became explicit (f2, top panel). Comets disappeared after ~100 s (bottom panel, t2), but returned after illumination was switched off for 30 s (t3). The process was repeatable after another 100 s illumination and then another light off period (t4 and t5). The alternate white and black bars illustrate the scheme of alternate illumination and dark periods. Scale bars, 5 μm.

DOI: https://doi.org/10.7554/eLife.26002.002

(hatched bars). The average velocities are signifantly different (Mann-Whitney U test, p<0.001), at 5.8 ± 0.26 and 7.9 ± 0.42 μm/sec respectively (*Figure 1e*), which are within the normal range measured in diverse eukaryotic cells (*Harris et al., 2016*).

Curiously, in some long recordings, comets suddenly gave way to a bird cage-like pattern (*Figure 1f1*, top panel; *Video 2* ) as if all cortical MTs were revealed by anti-tubulin immunofluorescence (*Horst et al., 1999*; *Dymek et al., 2006*). Comets returned automatically after ~1 min (middle and bottom panels). This unpredictability suggests that this pattern is caused by fluctuated intracellular conditions. When cells gradually flattened by the coverslip, MTs also became visible as a broken bird cage with comets (*Figure 1f2*, top panel, t1; *Video 3*). However, comets disappeared after ~100 s (bottom panel, t2). Interestingly, after a 30 s dark period, comets returned upon excitation light was switched back (t3). Simply alternating the dark and light period replicated the disappearance and return of comets (t4 and t5). These observations demonstrate that *Chlamydomonas* MT system is sensitive to certain signals; and suggests that excitation illumination creates a condition that is unfavorable for MT dynamics, but is reversed in the dark. As illumination opens channelrhodopsins that conduct a number of cations and Cl$^-$ (*Nagel et al., 2002*; *2003*); reviewed by *Hegemann and Berthold, 2009*), we hypothesize that changes of electrolyte concentrations modulate the MT system in *Chlamydomonas*. We test this on WT cells and mutant cells with more stable MTs under a number of treatments. Considering the light sensitivity, EB1-NG was imaged with wide field microscopy using minimal light intensity. Key results were summarized in *Table 1*.

## Sequential changes in the MT system elicited by a short acetic acid pulse and subsequent wash

We first performed the well-defined pH shock, recording EB1-NG signals in cells exposed to acetic acid (HA) in two complementary devices, perfusion chamber and diffusion chamber (*Figure 2a*). Recording of events in perfusion chambers started immediately before injection of 20 mM HA/TAP. Similar to pH shock, all cells were subjected to a swift change of environments as the injected solution was immediately pulled across the chamber by filter papers placed at the opposite end (*Wheeler et al., 2008*). However, the initial period was not decipherable due to flowing of injected fluid and unattached cells. For the diffusion chamber encircled by Vaseline, cells resuspended in the HEPES buffer were placed at one side, underneath the coverslip and objective lens (right panel). Recording started after 100 mM HA was injected into the opposite side and cells stopped flowing. This design that decelerated the acidification process captured the events as HA diffused across the field and into cells that were being recorded. However, the precise exposure time was uncertain; HA concentrations increased gradually, deflagellation was less effective; and not all cells were acidified to the same degree at the same time. TAP media were replaced with the HEPES buffer that only contained HEPES and Na$^+$, and was commonly used in *Chlamydomonas* experiments. Also, the enclosed chamber cannot be washed.

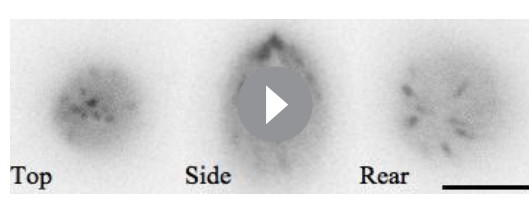

**Video 1.** (for *Figure 1b*) EB1-NG comets in WT cells.
DOI: https://doi.org/10.7554/eLife.26002.003

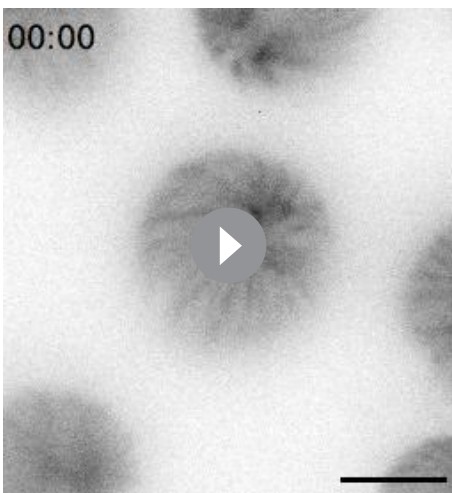

**Video 2.** (for *Figure 1f1*) Transient bird-cage pattern in WT cells that occurred sporadically during imaging.
DOI: https://doi.org/10.7554/eLife.26002.004

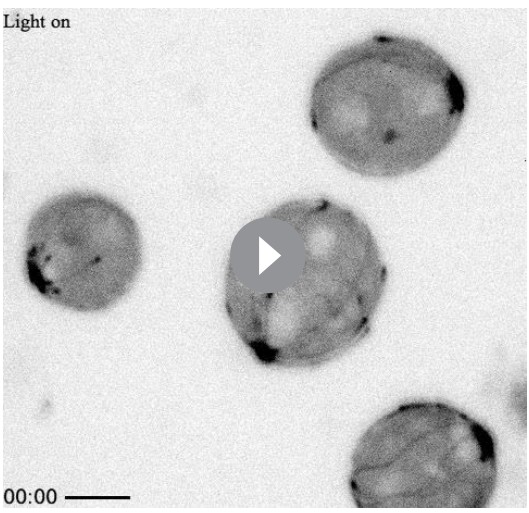

**Video 3.** (for *Figure 1f2*) Disappearance and return of comets in compressed cells following alternate periods of illumination and darkness.
DOI: https://doi.org/10.7554/eLife.26002.005

**Table 1.** Summary of the treatments and corresponding patterns of microtubules (MTs) and EB1-NeonGreen (NG) in wild type and *tub2* mutant cells.

Identical treatments were highlighted with a same gray shade for easy comparison. HA, acetic acid; TAP, Tris-Acetate-Phosphate culture medium.

| Treatments | MT (revealed by EB1-NG) | EB1-NG |
|---|---|---|
| *WT* | | |
| TAP or 5 mM $Na^+$/HEPES (Control) | Dynamic; cold labile | Comets |
| 20 mM pH4.5 HA/TAP pulse (perfusion as pH shock) | Invisible | No comets |
| 100 mM HA pulse (diffusion) | Shrink | Fibers; long comets |
| 10 mM pH3.0 HA | Invisible | |
| 7.5 mM pH3.4 HA | Shrink | Fibers; no comets |
| 5.0 mM pH3.5 HA | Dynamic | Fibers; comets |
| 10 mM pH3.0 HA bath; 5 mM $[Na^+]_{ex}$ wash | Freeze; bundle/branch; cold resistant | Fibers; no comets |
| 10 mM pH3.0 HA bath; 5 mM $[K^+]_{ex}$ wash | Dynamic | Comets |
| 21 mM $Na^+$/EGTA | Freeze; bundle/branch; cold resistant | Fibers; no comets |
| 21 mM $K^+$/EGTA | Dynamic | Comets |
| 55 mM $[Na^+]_{ex}$ | Dynamic | Long comets |
| 150 mM $[Na^+]_{ex}$ | Absence or a few low dynamic MTs | Fibers; no comets |
| TAP + 55 mM $[Na^+]_{ex}$ | Dynamic | Fibers; long comets |
| TAP + 100 mM $[Na^+]_{ex}$ | Dynamic | Fibers; long comets |
| TAP + 200 mM $[Na^+]_{ex}$ | Absence or a few low dynamic MTs | Fibers; no comets |
| *tub2* (colchicine-resistant) | | |
| TAP or 5 mM $Na^+$/HEPES | Dynamic | Comets |
| 10 mM pH3.0 HA | Freeze | Fibers; no comets |
| 10 mM pH3.0 HA bath; 5 mM $[Na]_{ex}$ wash | Freeze; bundle/branch | Fibers; no comets |

DOI: https://doi.org/10.7554/eLife.26002.006

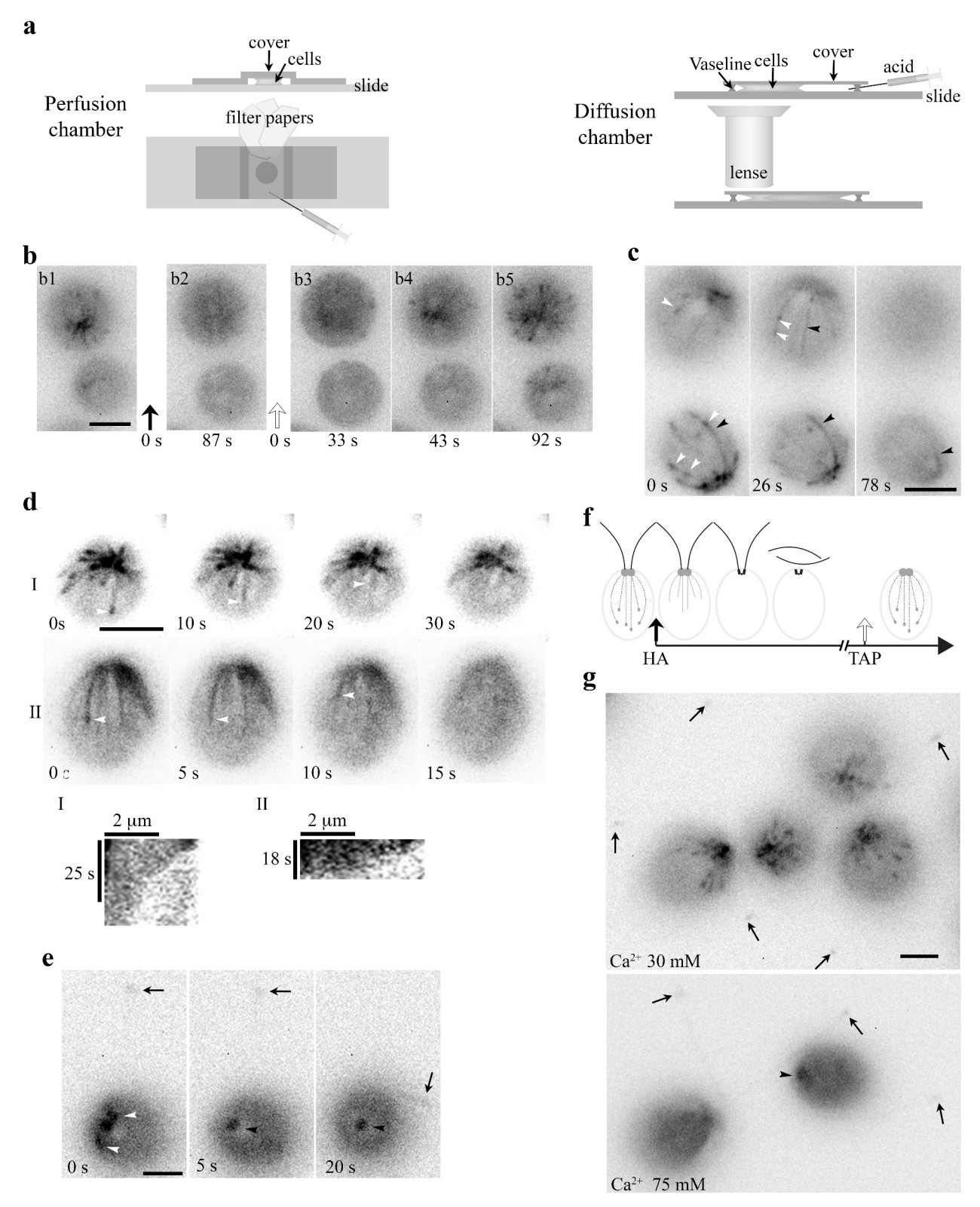

**Figure 2.** An HA pulse elicited swift sequential changes in the MT system. (a) Schematics depicting an open-ended perfusion chamber (left panels) and a diffusion chamber (right panels) for capturing the HA-induced rapid changes. (b) A 10 µl aliquot of cells resuspended in the TAP medium was placed in a perfusion chamber. The images (b1, 2) were captured before and after perfusion with 20 mM HA/TAP (pH4.5, t = 0, black arrow). The following recordings (b3-5) captured the events right after the TAP medium (pH7) was injected to wash away HA (t = 0, clear arrow). B3 is the first clear image
*Figure 2 continued on next page*

*Figure 2 continued*

after fluid and cells stopped flowing. Comets already disappeared within 87 s after HA perfusion. They started emerging 43 s after wash. (c) The process preceding HA-induced disappearance of EB1 comets in diffusion chambers. A 40 µl aliquot of cells resuspended in HEPES was placed in a diffusion chamber encircled by Vaseline, under the coverslip and an objective lens. HA was injected to the other side of the chamber and diffused toward cells that were being imaged. During the gradual acidification process, both comets (white arrowheads) and shank binding MTs (black arrowheads) were evident first and then both patterns vanished. (d) Time-lapse images and kymographs revealed endwise resorption of EB1-decorated MTs (white arrowheads). (e) Comets (white arrowheads) in the cell body vanished first before the excision of flagella (arrows). Following deflagellation, EB1 diffused away from the tip. EB1 signals remained at BBs but was static (black arrowhead). (f) A schematic depicting sequential changes in MTs upon exposure to HA and a subsequent wash with the TAP medium. Dotted lines with a comet, growing MTs. Dotted lines alone, shrinking MTs. Solid lines, shrinking MTs with EB1 shank binding. (g) Effects of $[Ca^{2+}]_{ex}$ on MTs. In cells resuspended in 30 mM $[Ca^{2+}]_{ex}$, flagella remained attached (arrows in left panel), while comets were vibrant. In the 75 mM $[Ca^{2+}]_{ex}$ group, comets disappeared and flagella were amputated (arrows in right panel). Static EB1 signals remained at BBs (arrowhead). Scale bars, 5 µm.

DOI: https://doi.org/10.7554/eLife.26002.007

Following the injection of 20 mM HA/TAP into the perfusion chamber, all comets vanished in the first discernable image taken at ~90 s (*Figure 2*, b1-b2). After wash with 10 mM pH7 HEPES, comets re-appeared at the BB area within ~45 s and MT dynamics resumed (b3-b5). Thus, HA exposure elicits the disappearance of EB patterns either by perturbing EB-MT interplays or causing cortical MTs to pause or disassemble. New dynamic MTs re-form rapidly after HA is washed away.

Diffusion chambers in which the acidification process occurred gradually allowed us to capture another unexpected phenomenon before comets vanish. A broken bird cage pattern with a few MTs (*Figure 2c*, left and middle panel, black arrowhead) and comets (white arrowheads) appeared before the disappearance at a time designated as 78 s (right panel). To decipher the disappearance, we analyzed digitally enhanced recordings (*Figure 2d*, top panels). As shown in two representative cells, overall EB1 signals were fading with time, which could be due to photobleaching, pH sensitivity of fluorescent proteins, disassociation of EB1 or system-wide MT disassembly. Some MTs clearly underwent endwise resorption (arrowheads). Kymographs tracking plus ends of prominent MTs in two rare still cells show that, as expected, comets disappear before the resorption of the respective MT. Measurements of the slopes show that the initial shortening speeds for these two events are 8 and 13 µm/min but decelerate toward the BB area. These speeds revealed by EB1-NG are in line with the averaged 10 ± 3 µm/min shortening speed of cortical MTs in cultured tobacco cells (*Dixit and Cyr, 2004*), faster than kinesin-13 catalyzed shortening (*Helenius et al., 2006*), and slower than the maximal shortening speed (30 µm/min) of MTs assembled from purified tubulins (*O'Brien et al., 1997*). Statistics of shortening speed was not analyzed because of few motionless cells and few shortening MTs with a definitive plus end. Tubulin reporters will be more appropriate for shortening analysis.

As for cells with EB1 signals detected in the cell body and flagella (*Figure 2e*) simultaneously, comets (white arrowheads in left panel) vanished first (middle panel) before flagella (arrow) were amputated (right panel). EB1 signals remained at the BB area but were static (black arrowhead). Contrary to deflagellation within seconds upon HA perfusion (*Wheeler et al., 2008*), the deflagellation in the diffusion chamber takes more than one minute due to gradual acidification. Thus, when cells are exposed to HA, shank binding increases, comets disappear, endwise resorption becomes evident and then flagella become amputated. These events do not require the contents unique to TAP media. Flagellar regeneration was not assessed because both chambers were not suitable for the long regeneration process. The sequential events occurring in the diffusion chamber are summarized in *Figure 2f*.

Lowering intracellular pH elicits $Ca^{2+}$ influx, whereas $Ca^{2+}$ prevents MT formation and promotes MT disassembly (*Weisenberg, 1972*; *O'Brien et al., 1997*). To differentiate whether HA-induced changes are due to $H^+$ or $Ca^{2+}$, we first raised $[Ca^{2+}]_{in}$ without adding HA. Calcium ionophore A23187 did not trigger deflagellation or evident changes in the MT system in our hands. This is not surprising since A23187 cannot elicit consistent effects in *Chlamydomonas* (*Bloodgood and Levin, 1983*). So we simply raised $[Ca^{2+}]_{ex}$. Perfusions of either HA or $CaCl_2$ solution elicit $Ca^{2+}$ influx, leading to deflagellation, although the latter is less efficient (*Wheeler et al., 2008*). WT cells resuspended in 30 mM $Ca^{2+}$/HEPES appeared agitated, suggesting entry of $Ca^{2+}$ (*Figure 2g*, top panel). However, flagella remained attached (arrows) and comet activity was robust. When cells were resuspended in 75 mM $Ca^{2+}$/HEPES solution (right panel), cells shed flagella (arrows). Comets already

disappeared, while static EB1 signals remained at the BB area (arrowhead). Thus, the outcomes elicited by high $[Ca^{2+}]_{ex}$ and $\geq$ 20 mM HA are similar.

## HA-induced phenomena with reduced concentrations and in a tubulin mutant

To dissect HA-induced phenomena, we treated WT cells with less concentrated HA. As shown in two top view images taken 40 s apart immediately after resuspension in 5 mM HA (*Figure 3a*; *Video 4*), all WT cells had motile flagella and had a dynamic, rather than still, bird cage pattern in which MTs shrink or grow with a comet, as if the entire MT system were revealed by fluorescent tubulins and fluorescent EB1 simultaneously. The side view images recorded 22 secs apart revealed endwise resorption of a shrinking MT (arrowhead). Only a few resorbing MTs were captured in cells resuspended in 7.5 mM HA (*Figure 3a*, right panel). This still cell allows us to plot the kymograph, which shows a tapered endwise resorption with the initial shortening speed of 4 mm/min. All MT patterns and comets disappeared in cells resuspended in 10 mM HA. Therefore, low [HA] increases EB1 shank-binding, rendering the bird-cage pattern. As HA concentrations increase, MTs stop growing and comets are lost. As resorption continues, perhaps even at a hastening pace, comet and bird cage patterns vanish. The 5 mM HA experiment partially replicates the light-induced sporadic transient appearance of the bird cage pattern in WT cells (*Figure 1*, f1). Changes elicited by 7.5 mM HA partially mimics HA-induced responses in diffusion chambers (*Figure 2b-e*).

Independently, we took a genetic approach, testing a tubulin mutant, *tub2*, hypothesizing that HA-elicited resorption would decelerate due to the more stable MTs resulted from a missense mutation in β-tubulin (*Schibler and Huang, 1991*). In the EB1-NG transgenic *tub2* cells, the comet pattern (*Figure 3b*, top panels) appeared indistinguishable from that in WT transgenic cells (*Figure 1b*).

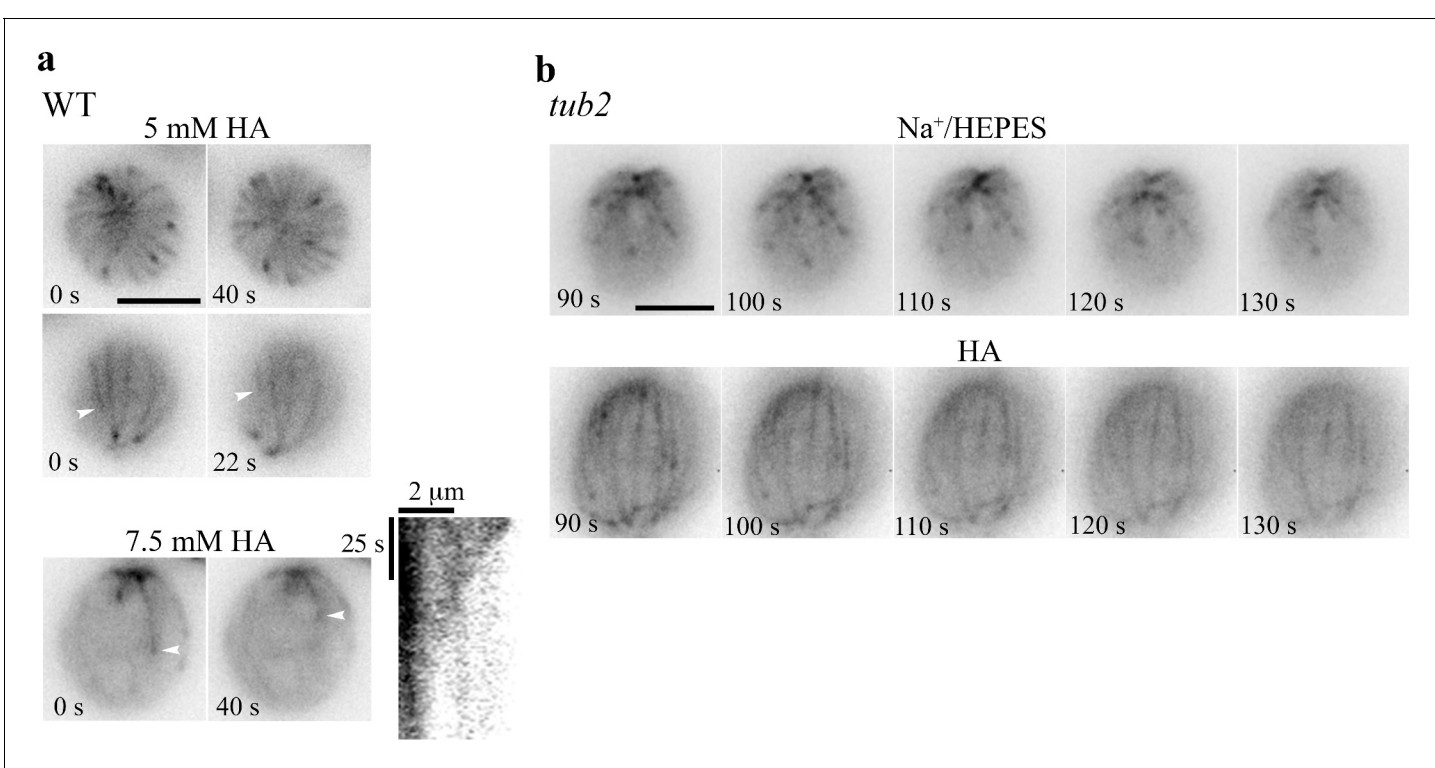

**Figure 3.** Abatements of HA-induced changes. (a) Concentration-dependent HA effects on MTs in WT cells. WT cells resuspended in 5 mM HA exhibited a bird cage pattern with dynamic comets (top and middle panels). Shrinking MTs were visible in the two side view images taken 22 s apart (white arrowhead). Most patterns were absent in cells resuspended in 7.5 mM HA, except a few shrinking MTs (bottom panels, white arrowheads). A kymograph revealed the endwise resorption. (b) EB1-NG in *tub2* cells appeared as normal comets in pH7.4 Na$^+$/HEPES (top panels), but as a bird cage with fine MT fibers in 10 mM pH3 HA (bottom panels). Most MTs froze, but some appeared in and out of focus. Time-lapse images were shown at 10 s intervals. Scale bars, 5 μm.
DOI: https://doi.org/10.7554/eLife.26002.008

Thus, our assay is not sensitive enough to report the increased MT stability. Interestingly though, in the image taken immediately after resuspension in 10 mM HA, the bird cage pattern with a few comet-like spots already formed in every *tub2* cell (bottom panels). The pattern was rather stable, although some MTs seemed to be out of focus intermittently (bottom right panel), suggesting detachment from the plasma membrane. Thus, HA-induced diminished EB1-NG signals in WT cells are not due to reduced brightness of NG at low pH (*Shaner et al., 2013*).

Collectively, these data show that, in ~5 mM HA, dissociation of EB1 from MT plus ends is inhibited, leading to a bird cage like pattern. More concentrated HA blocks MT elongation while accelerates MT depolymerization, which is prohibited by increased MT stability in *tub2* cells. Chelation of $Ca^{2+}$ by pretreatment of cells with buffered 10 mM EGTA did not hinder these changes, whereas pH3, 10 mM HCl did not elicit any responses. The simplest interpretation is that, as with deflagellation, HA-elicited changes in cortical MTs are due to intracellular rather than extracellular acidification; however, the changes do not require extracellular $Ca^{2+}$.

## Formation of cold-resistant thick MT fibers in WT cells recovered from HA bath

Intracellular pH is tightly controlled. *Chlamydomonas* expresses $Na^+/H^+$ exchangers (*Pittman et al., 2009*), as well as various channels and pumps for different locations (*Fujiu et al., 2011*; *Taylor et al., 2012*) to maintain electrolyte homeostasis. To exacerbate electrolyte imbalance, we extended HA exposure, assuming that accumulated $H^+$ ions from a lengthy exposure would be replaced with other cations. As expected, EB1 patterns were absent except for the static signal at the BB area in cells resuspending in pH3, 10 mM HA/double distilled water (ddw) for 5 min (*Figure 4a*, left panel). After wash with $Na^+$/HEPES in a perfusion chamber, dynamic EB1 signals emerged at the BB area after ~50 s (*Figure 4a*, cell I in right panel). Interestingly, nascent MTs were not adorned with a typical comet. Compared to the bird cage pattern, they appeared thicker, fewer and nearly uniformly decorated, as if plus end tracking EB1 stayed behind growing MTs. In cells recorded at a later period (cell II and III, between 60–159 s; *Video 5*), MT growth decelerated, especially between 109–159 s. The late events became unclear due to photobleaching from continuous illumination. In the recordings started 180 s after wash, all dynamic processes stopped (*Figure 4b*, left panel), as illustrated by two nearly identical images taken 20 s apart of two representative cells (cell I and II, right panels). Notably, some static fibers split, or had more than one comet aligned in tandem (arrowheads). Taken together, these observations strongly suggest that nascent MTs generated after HA bath and wash are abnormal, perhaps with a propensity to nucleate ectopically, branching or growing on top of the other piggy-backing as bundles. Kymographs comparing three representative MTs (*Figure 4c*, top panels) with growing MTs in the untreated control cells (bottom panels) confirm that EB1 signals reach near the BB area in cells recovering from HA bath. MT growth rates, shown by the slopes, fluctuate but are mostly slower than those in untreated control cells. Overall the rates decline until MT growth stops. Thus, compared to the numerous, thin, dim MT fibers in the transient bird cage pattern (*Figure 3a*), MTs formed in the recovery phase after HA bath were fewer, thicker, shorter, brighter and long-lived.

Dynamic MTs are cold labile. Cold treatment induces MT endwise resorption in vitro (*Müller-Reichert et al., 1998*). To test the stability of these thick frozen MTs formed in the recovery phase, glass slides with a droplet of cells after HA bath and wash were placed on ice for 3 min. Images were taken using the microscope at the room temperature immediately, about 20 s after slides were removed from ice. Unexpectedly, most cells were imaged from the apical end and the focal planes drifted continuously, indicating that cells oriented toward the objective lens and floated gradually during this warm up period. As shown in two representative cells, EB1-decorated MTs after HA bath and wash remained after the removal from ice amidst the drift of focal planes (*Figure 4d*, top panels). Thus, the thick static MTs formed after HA bath and wash are cold stable. The pattern remained for the subsequent 70 s recording period. In contrast, for the control without HA bath, EB1 patterns were undetectable initially (bottom panels), indicating cold lability. Dynamic EB1 signals gradually re-emerged at BBs after 30 s (white arrowheads).

To learn if frozen thick MTs after HA bath and wash were reversible, we continued imaging cells in the recovery phase. To prepare for the long recovery, wash buffer was replaced with TAP media. Thawing signs emerged gradually. Comet (*Figure 4e*, white arrowheads) activity was vibrant in cells

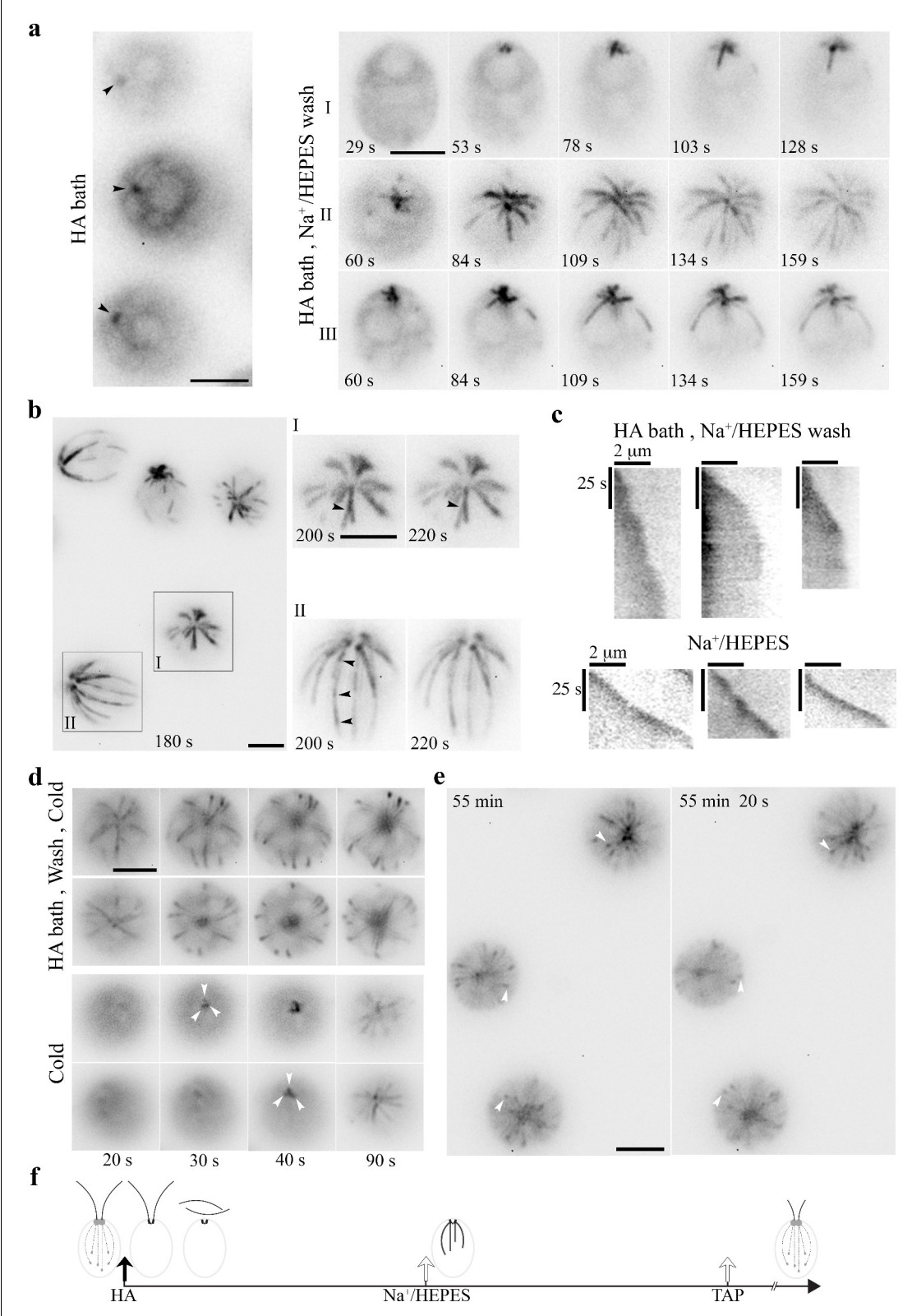

**Figure 4.** HA bath and a subsequent wash induced long-lived yet reversible changes to the MT system. (a) Static EB1 signals remained at BBs after cells were resuspended in 10 mM pH3 HA for 5 min (left panel). After replacing HA with the HEPES buffer, EB1 signal at the BB area intensified within 1 min (black arrowheads). But newly formed MTs were thick and prominent, lacking the typical comet (right panel, cell I). In cells recorded 60 s after wash (cell II and III), MT elongation slowed down gradually. (b) In cells imaged ~180 s after wash, EB1-decorated MTs in all cells stopped growing (top panel),

*Figure 4 continued on next page*

*Figure 4 continued*

DOI: https://doi.org/10.7554/eLife.26002.010

*Figure 4 continued*

as highlighted in two additional images of two representative cells captured 20 s apart. In addition, a MT fiber in cell I split into two (arrowhead), while a fiber in cell II had multiple comets aligned in tandem (arrowheads), as if new MTs nucleated or grew on older ones. (c) Kymographs comparing the growth of three representative MTs in cells pretreated with 5 min HA bath and then the wash buffer (top panels) and in control cells in the HEPES wash buffer (lower panels). Comets in the latter manifested as an intense spot at the plus end. The sharper slopes in the former indicated slower growth. Unchanged slopes indicated paused growth. (d) Long-lived MTs formed after HA bath and wash were cold resistant (top panels). As shown in two representative cells, frozen MTs remained in the image captured 20 s after 3 min on ice. The changes in subsequent images were due to focal plane drifts as cells were floating gradually as warming up. In control cells without previous HA exposure (bottom panels), EB1 signals were absent initially (20 s), consistent with cold lability. Dynamic comets (white arrowheads) gradually emerged at the BB area afterwards. (e) Although the MT system froze within minutes after HA bath and HEPES wash (t = 0 min), dynamic comets resumed after 55 mins in cells recovered in TAP media as shown in two images captured 20 s apart. Scale bars, 5 μm. (f) A schematic summarizing the sequence of MT changes induced by HA bath and wash.

55 mins post wash, as if the freeze had never occurred. The sequential events occurring after HA bath and wash are summarized in *Figure 4f*.

## Distinct cold-resistant thick MT fibers in tub2 cells recovered from HA bath

We also tested how HA bath and wash affected *tub2* cells (*Figure 3*). As expected, the frozen bird cage pattern remained after 5 min HA bath (*Figure 5a*; *Figure 5—figure supplement 1*). After wash, comets returned (panel I and II) and lengthened as in WT cells. However, comets (white arrowheads) appeared at the plus end of existing MTs first rather than at BBs where nascent MTs emerged in WT cells (*Figure 4a*). The bird cage pattern gradually faded concomitantly. For images taken at 60 s and later after wash, comets formed and moved along existing MTs (Panel III, *Video 6*). This strengthens the interpreted bundling propensity of MTs in WT cells recovered from HA bath (*Figure 4b*). The responses of *tub2* cells to HA and wash are summarized in *Figure 5b* and *Table 1*. These collective results from *tub2* indicate that 10 mM HA cannot elicit the resorption of the more stable MTs in *tub2*, as such the bird cage pattern with cortical MTs uniformly decorated with EB1 persists; the other reactions are similar in *tub2* and WT cells recovering from HA bath, but comets preferentially return to old MTs, which need to resorb to make room for new ones to maintain the number of cortical MTs.

## The long-lived EB1-decorated MTs after HA bath and wash are due to the rise of intracellular [Na$^+$] but not [K$^+$]

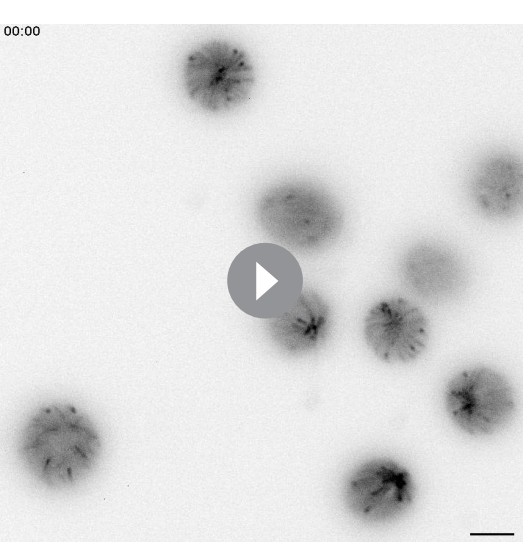

To identify the ion causing the formation of static thick EB1-decorated MT bundles in cells recovering from HA bath, HA-bathed cells were washed with different solutions. Interestingly, as shown by time-lapse images separated by a 10 s interval, EB1-decorated MT fibers formed only if the wash solution contained Na$^+$, such as the NaOH-buffered HEPES (5 mM Na$^+$) or 5 mM NaCl in ddw (*Figure 6a*, top two panels). On the other hand, comets resumed profusely if the wash solution lacked Na$^+$, such as the KOH-buffered HEPES buffer, 5 mM KCl in ddw or plain ddw (bottom three panels). Therefore, Na$^+$ accounts for the reformation of thick, long-lived static MTs in the recovery phase of HA-bathed cells.

Since Na$^+$ has low permeability compared to K$^+$ (*Ronkin and Buretz, 1960*), we predict that Na$^+$ ions from the wash buffer are entering the cytosol when Na$^+$/H$^+$ exchangers are removing H$^+$. To test this, we resuspended WT cells in 10

**Video 4.** (for *Figure 3a*) WT cells treated with 5 mM HA.
DOI: https://doi.org/10.7554/eLife.26002.009

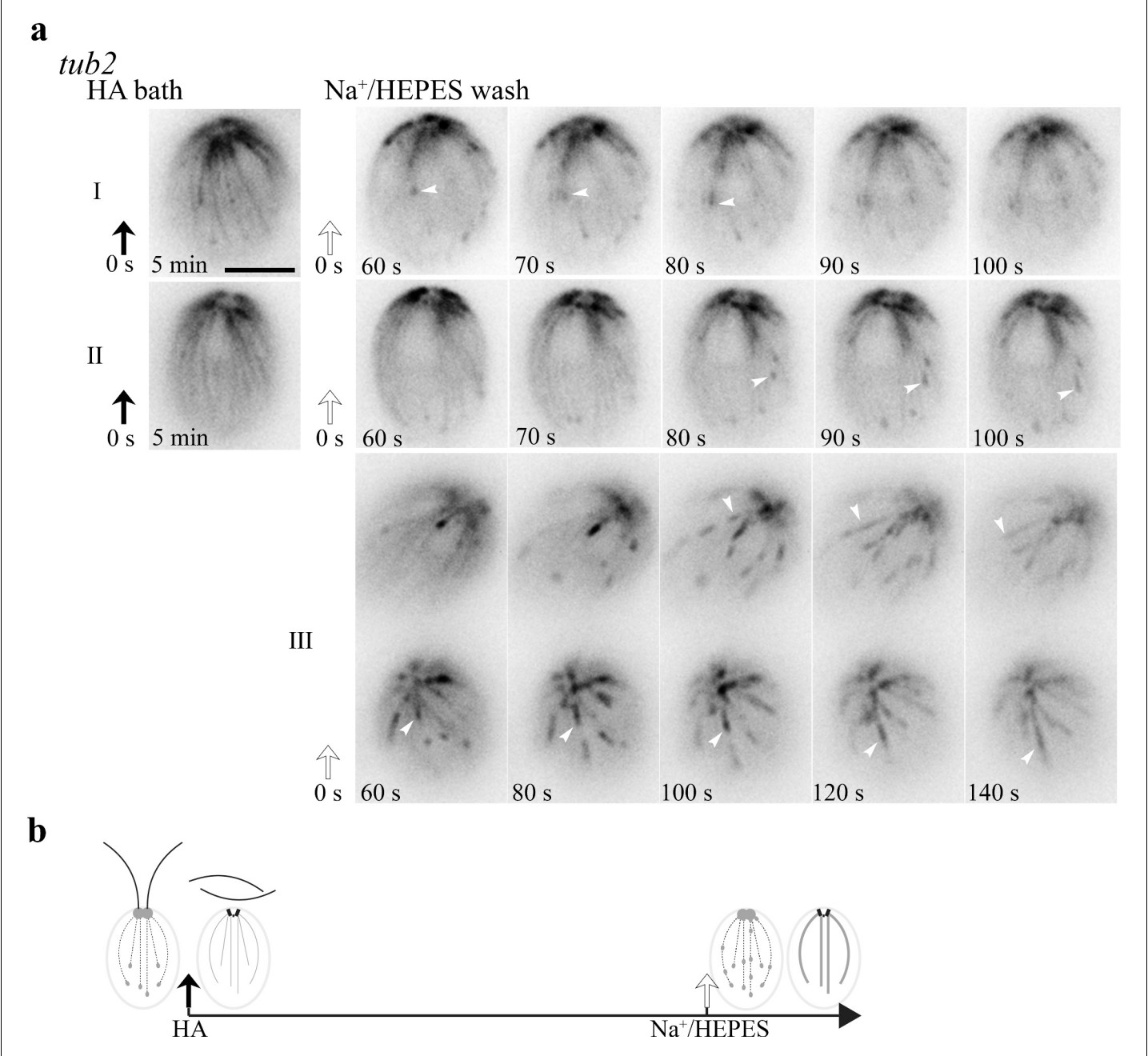

**Figure 5.** The MT pattern in *tub2* cells recovered from the HA bath. (**a**) The frozen bird cage pattern of fine MTs remains after 5 min HA bath as shown in two representative cells (left panel). After wash, dynamics resumes within 60 s. Notably, comets (panel I and II, white arrowheads) reemerged at the plus end of existing fine MTs, instead of the BB area. Meanwhile the bird cage pattern was fading. MTs were growing, but comets were lengthening and the pace was very slow. In videos recorded only after wash, the recovery was faster. Some comets already emerged from the BB area. Some moved along old MTs and lengthened (panel III, white arrowheads). (**b**) A schematic summary of HA-induced changes in *tub2*. Scale bars, 5 μm.

DOI: https://doi.org/10.7554/eLife.26002.012

The following figure supplement is available for figure 5:

**Figure supplement 1.** (for **Figure 5a**) *tub2* cells in 5 min HA bath.

DOI: https://doi.org/10.7554/eLife.26002.013

mM, pH8 EGTA buffered by NaOH (final ~21 mM Na$^+$). Concentrated EGTA and EDTA are classical tools for creating permeable cell models perhaps by Ca$^{2+}$ chelation or other mechanisms

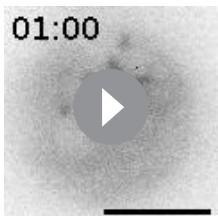

**Video 5.** (for *Figure 4a*) WT cells in Na⁺/HEPES after HA bath.
DOI: https://doi.org/10.7554/eLife.26002.011

(*Miller, 1979*; *Tazawa and Shimmen, 1983*; *Arikawa and Suzaki, 2002*; *Prachayasittikul et al., 2007*). However, EGTA at this concentration does not influence polymerization of MTs from purified tubulins in vitro (*Olmsted and Borisy, 1975*). As expected, thick EB1-decorated MTs formed after cells were resuspended in Na⁺/EGTA for 5 min (*Figure 6b*, left panels). Thick MTs appeared static after 10 mins and were cold-resistant. In contrast, cells resuspended in KOH-buffered EGTA had vibrant comet activities (right panels). However, cells quickly burst, indicating the perturbed plasma membrane. Given the effect of Na⁺ but not K⁺ on the MT cytoskeleton, EGTA solutions used in the other experiments were buffered by KOH.

To simplify the treatment, we tested several concentrations of NaCl. Compared to the growing MTs with typical comets in control cells resuspended in the HEPES buffer, in HEPES with 55 mM NaCl, MTs were still growing but with longer comets as shown in fluorescent imaging (*Figure 6c* panel I) and modest tapering of comet intensity in the linescan plot (panel II). The distribution of comet velocity (panel III) and the average velocity (panel IV) showed that $[Na^+]_{in}$ at this level only slightly reduced MT growth rate (two-tailed p-value=0.0497, <0.05). Therefore, high $[Na^+]_{ex}$ still could raise $[Na^+]_{in}$, increasing the time EB1 spent at the plus end and thus comet lengths. MT growth is also sensitive to $[Na^+]$ but less than EB1 plus end binding. For cells resuspended in 150 and 200 mM $[Na^+]_{ex}$ for 5 min, flagellar motility ceased. In most cells, comets vanished, although static EB1 signals remained at the BB area (*Figure 6d*), as in 75 mM $[Ca^{2+}]$ solution (*Figure 2g*). However, some cells still contain thick fibers (top panel), a signature of Na⁺-induced changes.

Together, these results show that the MT system undergo a range of changes as a function of $[Na^+]$. Frozen thick MTs caused by 21 mM Na+/EGTA or by 5 mM Na⁺/HEPES wash following an HA bath or may be triggered by less than 21 mM $[Na^+]_{in}$ respectively due to passive diffusion through the EGTA-permeabilized membrane or by H⁺/Na⁺ exchange. $[Na^+]_{in}$ is likely even lower in cells which are simply exposed to 55 mM $[Na^+]_{ex}$, and have long comets and growing MTs. In contrary, few or no EB1-decorated thick MT fibers caused by 150–200 mM $[Na^+]_{ex}$ or 75 mM $[Ca^{2+}]_{ex}$ suggest that MTs disassemble at the concentrations that deem hypertonic for *Chlamydomonas*.

## Correlations of Na⁺-induced changes in the MT system and cell division

Salt stress triggers an array of responses in *Chlamydomonas*, including inhibited multiplication (*Perrineau et al., 2014*; *Takouridis et al., 2015*). The consequence of Na⁺-induced changes in the MT system (*Figure 6*) could be assessed by cell division that requires a dynamic MT system. In typical synchronized liquid cultures, mother cells undergo 1–3 rounds of cell division during the dark period, each forming a sporangium with multiple daughter cells enclosed in the mother cell wall. Daughter cells are released before the light period (*Kubo et al., 2009*). To test this, aliquots of cell cultures were harvested at 8 PM right before the dark period. After resuspension in fresh media with increasing concentrations of NaCl, all tubes were kept in the dark until the light period. Sporangia and single cells were counted every two hours until 8 AM and at 5 PM. Sporangia ratios

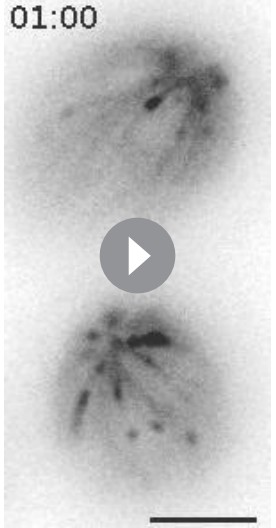

**Video 6.** (for *Figure 5a*) *tub2* cells in Na⁺/HEPES after HA bath.
DOI: https://doi.org/10.7554/eLife.26002.014

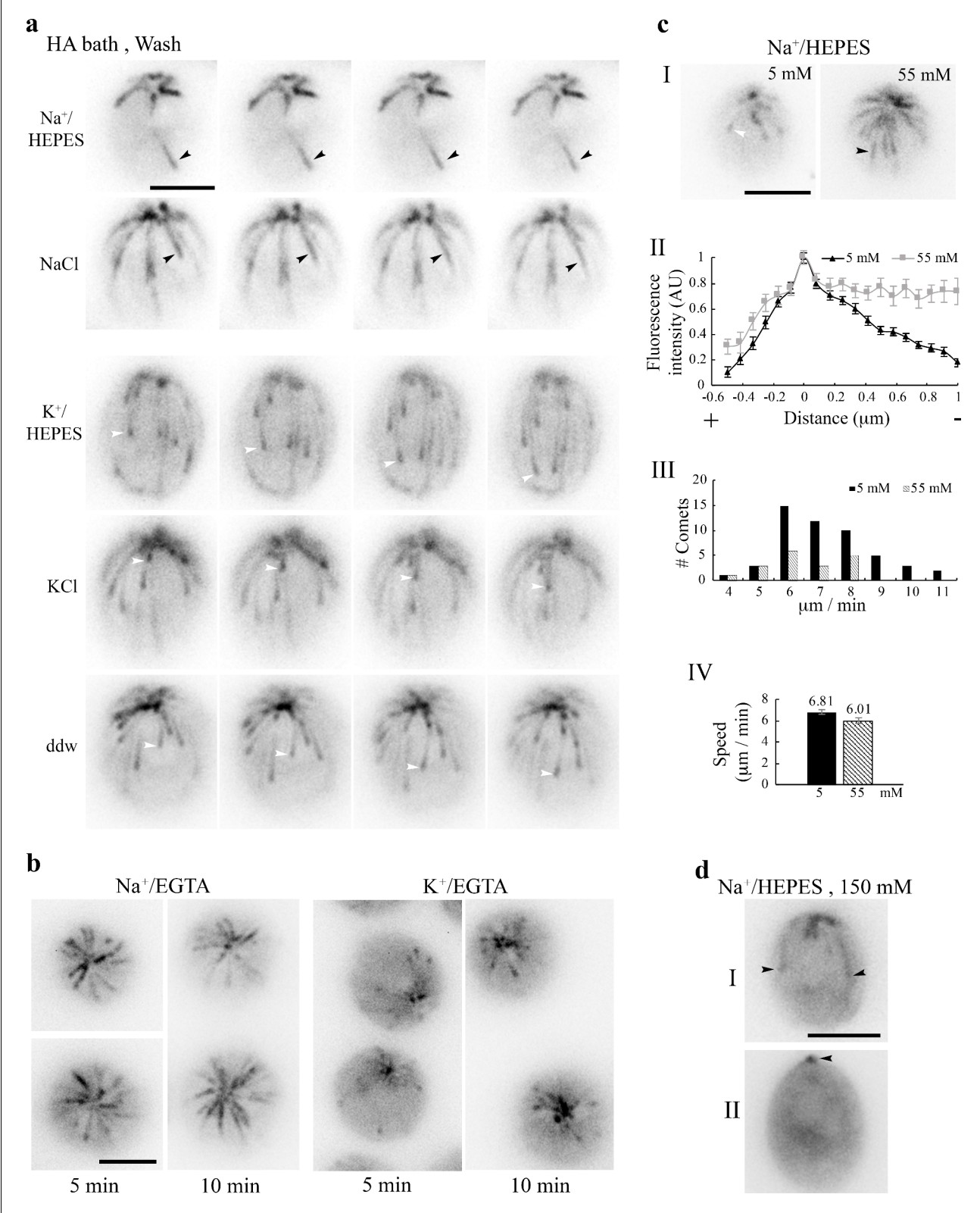

**Figure 6.** Na+-dependent changes of the MT system. (a) MTs in cells were largely frozen after 5 min 10 mM pH3 HA bath and 3 min in the wash solution, such as 5 mM pH7.4 Na+/HEPES buffer or 5 mM NaCl solution (black arrowheads). In contrast, growing MTs with a comet (white arrowheads) returned if the wash buffer lacked Na+, such as 5 mM K+/HEPES buffer, 5 mM KCl solution, or ddw. (b) Thick MTs in cells resuspended in 21 mM Na+/EGTA for 5 min or 10 min (left panel), contrary to comets in cells in 21 mM K+/EGTA (right panel). Thick MTs were still growing after 5 min incubation

*Figure 6 continued on next page*

*Figure 6 continued*

but static after 10 min incubation. (c) High $[Na^+]_{ex}$, without pre-exposure to HA, was sufficient to alter comet patterns. Contrary to typical comets in cells resuspended in the HEPES buffer with 5 mM $Na^+$, long comets were thick in cells resuspended in 55 mM $Na^+$ for 5 min (panel I). Normalized linescans confirmed little tapered intensity (panel II, n = 36 comets from 11 cells in 5 mM $Na^+$; n = 13 comets from 4 cells in 55 mM $Na^+$). As shown in the range of speed (panel III), long comets were moving, and the mean speeds of short and long comets were significantly different (panel IV, n = 51 from 11 cells in 5 mM $Na^+$; n = 18 from 4 cells in 55 mM $Na^+$) (p<0.05). (d) Two representative cells after 5 min in 150 mM $Na^+$/HEPES. Some cells still retained a few thick MTs (cell I). Some only had static EB1 signals at the BB area (cell II, arrowhead). Scale bars, 5 μm.

DOI: https://doi.org/10.7554/eLife.26002.015

were inversely related to NaCl concentrations (*Figure 7a*). The peak period was also delayed for 100 mM samples. No sporangia were observed in 200 mM samples. A repeated experiment for fluorescence imaging confirmed the trend, and showed no comets or EB1-decorated MTs in 200 mM samples (*Figure 7b*). As in 55 mM $Na^+$/HEPES (*Figure 6c*), comets lengthened in 55 and 100 mM $Na^+$/ TAP, but a linescan plot (*Figure 7c*, left panel) showed that the lengthening degrees appeared similar. Mean comet speeds (right panel), 8.38 ± 0.345 μm/min and 7.79 ± 0.628 μm/min respectively for 55 and 100 mM samples, were not significantly different (two-tailed t test, p=0.385), but faster than 6.41 ± 0.241 μm/min for 0 mM samples (two-tailed t test, p<0.001 for 0 mM and 55 mM samples; Mann-Whitney U test, p=0.05 for 0 mM and 100 mM samples). Thus, absence of comets and decorated MTs is consistent with blocked proliferation of cells exposed to 200 mM NaCl. However, the similar comet lengths and comet speeds for the 55 and 100 mM NaCl group do not correlate with differences in the numbers of dividing cells and the delay. We did not test whether cell division was inhibited when MTs froze (*Figures 4a–b* and *6a–b*), since the effects from HA bath- or EDTA-assisted increase in $[Na^+]_{in}$ last only one hour (*Figure 4e*) or perturbs the plasma membrane respectively. The HA treatment that induced bird cage pattern (*Figure 3a*) was not compatible with this experiment either, since prolonge acidification will result in programmed cell death (*Zuo et al., 2012*).

## Discussion

Expression of EB1-NG as a reporter loosens the bottle neck posed by autofluorescence of *Chlamydomonas* and unleashes the potential of this MT model system. Contrary to the perceived stability of the MT system in typical interphase animal cells (*Lieuvin et al., 1994*), EB1-NG reports in real time the various changes of algal MT system that are elicited by excitation illumination, compression, $H^+$ and $Na^+$. We summarize the changes and discuss possible underlying mechanisms; and the new insight on the MT system, plant salinity responses and additional concerns of environmental stresses.

### Changes elicited by $H^+$

The changes of algal MT system elicited by intracellular acidification are swift, stunning and novel (*Figures 2b–e* and *3*). Among heightened shank binding, diminished comet activity, paused MT growths, and MT shortening, the bird cage pattern of shank binding is the most sensitive, elicited reliably by 5 mM HA (*Figure 3a*). The recovery occurs either within a minute (*Figure 2b*) or protractedly (*Figure 4e*), depending on the exposure duration and wash solutions. Extracellular $Ca^{2+}$ is not required for the HA-induced changes, since they still occur after $K^+$/EGTA treatment. However, we cannot rule out the involvement of $Ca^{2+}$ released from intracellular storages and other signaling pathways.

Although the pH of $[HA]_{ex}$ that triggers these changes is ~3.4, we expect that the resulting intracellular pH is higher than 6.3. Firstly, HA-induced changes in diffusion chambers appear before deflagellation (*Figure 2e*) that occurs at pH6.3 (*Wheeler et al., 2008*; *Braun and Hegemann, 1999*). Secondly, the swiftness and reversibility (*Figures 2b, 3a, 4a* and *5a*) indicate that these changes are not triggered by apoptotic pathways, commonly induced by a drop of 0.3–0.4 unit from normal intracellular pH (*Lagadic-Gossmann et al., 2004*), which is 6.8 for *Chlamydomonas* (*Wheeler et al., 2008*). Furthermore, the bird cage pattern could be transiently triggered merely by illumination (*Figure 1f1*) that may open $H^+$-selective channels. Conversely, comets return to compressed cells when illumination is turned off for 30 s (*Figure 1f2*); or return within ~45 s after HA is washed away

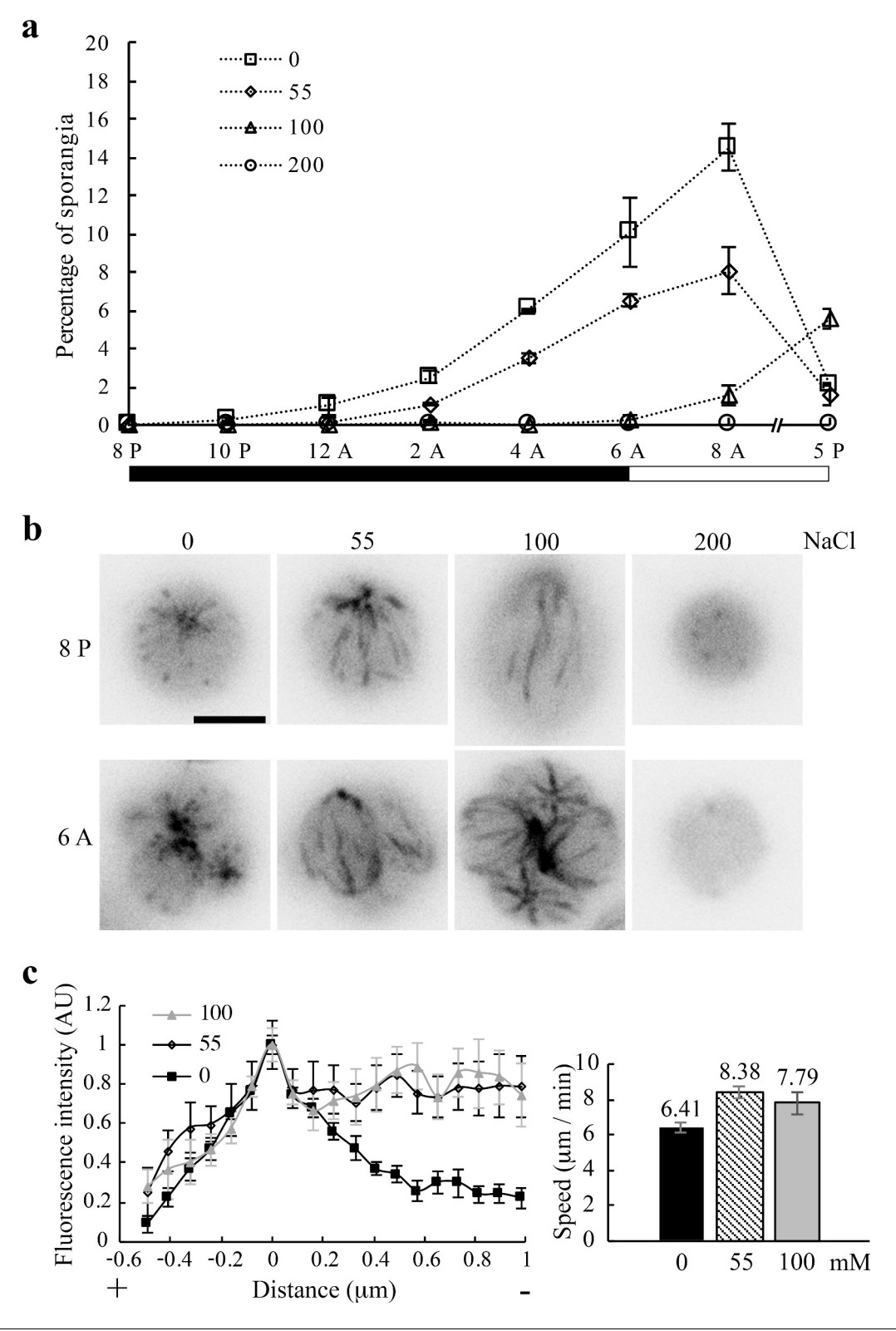

**Figure 7.** Na$^+$ concentration-dependent inhibition of cell division. (a) Reduced abundance and delayed appearance of sporangia with newly divided daughter cells as a function of NaCl concentration in TAP media. Triplicate cell pellets from a typical log-phase culture were suspended in each indicated solution immediately before the dark period. One aliquot from each sample was fixed with 2% glutaraldehyde periodically for 12 hr and

*Figure 7 continued on next page*

*Figure 7 continued*

then at 5 PM. (n > 500 single cells and sporangia). Black bar, dark period; clear bar, light period. (b) Representative single cells and sporangia for each [NaCl] group immediately before and after the dark period respectively. Comets lengthened in 55 mM and 100 mM $Na^+$/TAP, while MTs remained dynamic in cells at both states. No comet or sporangium was evident in 200 mM samples. Scale bar, 5 μm. (c) Quantifications of comets. A linescan plot of comets in 8 PM samples (left panel) showed longer comets in cells in the 55 mM (n = 8 comets from 3 cells at 55 mM) and 100 mM group (n = 7 comets from 2 cells) than that in the the 0 mM group (n = 13 comets from 8 cells). Mean comet speeds (right panel) in the 55 mM (n = 24 from 5 cells) and 100 mM group (n = 10 from 2 cells) taken at 10 PM were significantly faster than that in the 0 mM group (n = 42 from 7 cells) (two-tailed t test, p<0.001 for the 0 mM and 55 mM group; Mann-Whitney U test, p=0.05 for the 0 mM and 100 mM group). The difference of the 55 mM and 100 mM group was not statistically significant (two-tailed t test, p=0.385).
DOI: https://doi.org/10.7554/eLife.26002.016

(*Figure 2b*). These observations strongly suggest that a slight imbalance of pH homeostasis is sufficient to elicit immediate changes in algal MT system.

Although pH affects proteins' ionization and thus their functions and protein-protein interactions in general (*Hepler, 2016*), we speculate that the MT system is particularly sensitive to declining pH because of the acidic isoelectric points of tubulins and EB1. For example, the respective pI of *Chlamydomonas*α-tubulin, β-tubulin and EB1 is 5.01, 4.82, and 5.7. A decrease of pH from the resting level will make these proteins less negatively charged, especially at their C-terminal acidic tails that are central to MT-accessory protein interplays and the targets of various post-translational modifications (reviewed by *Song and Brady, 2015*; *Buey et al., 2011*; *Rovini et al., 2013*). The resulting decreased repulsion could explain increased affinity of EB1 to GDP-tubulin, leading to nearly immediate appearance of the bird cage pattern in WT cells exposed to 5 mM HA (*Figure 3a*) or in *tub2* cells exposed to 10 mM HA (*Figure 3b* and *5a*). As pH decreases further, additional changes in protein conformation may inhibit the growth of MTs and EB1 binding to plus ends, leading to comet reduction or ultimate disappearance.

## Changes elicited by $Na^+$

The responses elicited by slight increases in $[Na^+]_{in}$ and $[H^+]_{in}$ are distinct in pattern and pace. Contrary to instant appearance of $H^+$-elicited bird cage pattern with fine, individual MTs (*Figure 4d*), as $Na^+$ continues rising, comets lengthen; cortical MTs undergo ectopic nucleation, splitting, bundling, decelerate and stop eventually (*Figures 4–6*). The slower pace of low-$Na^+$-induced responses are likely due to low $Na^+$ permeability. The direct switch from lengthening comets to MT resorption (*Figures 6c*, *7b and c*) suggests that NaCl alone cannot raise $[Na^+]_{in}$ incrementally. Likewise, $Ca^{2+}$ permeability is tightly controlled. As such $[Na^+]_{in}$ and $[Ca^{2+}]_{in}$ cannot be adjusted as nimbly as $[H^+]_{in}$ with permeable HA.

Continual aggravated changes until MTs become static during a 10 min period in 21 mM $Na^+$/EGTA (*Figure 6b*) indicate that they are occurring before $[Na^+]_{in}$ reaches 21 mM, which is made possible by EGTA treatment. Similar $Na^+$-dependent responses emerge faster following a HA bath – in 45–180 s in various wash solutions containing only 5 mM $Na^+$ (*Figures 4a* and *6a*), suggesting accelerated rise of $[Na^+]_{in}$ due to the activity of $Na^+$/ $H^+$ exchangers to remove accumulated $H^+$ ions (*Pittman et al., 2009*). On the other hand, by simply relying on limited passive diffusion through the normal plasma membrane, the rise in $[Na^+]$ may be the most modest in 55–100 mM $[Na^+]_{ex}$ that is sufficient to increase comet length but only slows down MT growth rate slightly (*Figures 6c*, *7b and c*). Based on the incremented responses, rather than all-or-none responses to a threshold, we speculate that algal MT system changes as a function of $[Na^+]_{in}$. Contrary to $[Na^+]_{ex}$, raising $[K^+]_{ex}$ had no evident effect (*Figure 6a*). This is reasonable, given high $[K^+]_{in}$, ~70 mM in *Chlamydomonas* cells (*Malhotra and Glass, 1995*). This highlights the selective sensitivity of the algal MT system to $Na^+$ and rules out mere ionic effects. One interesting possibility is that $Na^+$ binds to particular sites in algal tubulins, analogous to $Ca^{2+}$ binding sites in mammalian tubulins (*Solomon, 1977*; *Serrano et al., 1986*).

## Common changes elicited by high extracellular HA, Na⁺ and Ca²⁺

EB1 signals largely vanish at or above 150 mM $[Na^+]_{ex}$ (*Figures 6d* and *7b*), as in 10 mM HA (*Figures 2c* and *4a*) and 75 mM $[Ca^{2+}]_{ex}$ (*Figure 2g*) except residual static signals at the BB area. These changes likely reflect simultaneous disassembly and paused new growth, and an immobile EB1 population underneath BBs respectively (*Yan et al., 2006*; *Pedersen et al., 2003*). The similar outcomes caused by distinct ions and obvious shrinkage of the cell body at even higher concentrations of Na⁺ and Ca²⁺ suggest that hypertonicity is involved. We envisage that high concentration responses could be caused by one cation exceeding a threshold concentration; or concurrent rises of multiple electrolytes as $H_2O$ moves out of cells.

The capture of endwise resorption only in low concentration conditions (*Figures 2d* and *3a*) suggests that increased concentrations of these ions heighten shortening- the incidence and/or speed. This mimics the high Ca²⁺ effects in vitro. Ca²⁺ blocks MT formation (*Weisenberg, 1972*), whereas 0.5–0.6 mM Ca²⁺ - in the absence of MAPs - could increase shortening incidence and accelerate shortening speed of MTs beyond 150 µm/min (*Karr et al., 1980*; *O'Brien et al., 1997*). Although pH shock and high $[Ca^{2+}]_{ex}$ only temporarily raise $[Ca^{2+}]_{in}$ up to 1 µM (*Wheeler et al., 2008*), the lower concentration may be sufficient to accelerate shortening.

Some electrolyte-elicited changes could be evoked by signaling pathways in *Chlamydomonas* as in other organisms (*Wang et al., 2013*; *Perrineau et al., 2014*; *Khona et al., 2016*), which perhaps occur later and slower; and last longer than the direct effects from electrolytes. For example, kinesin-13 becomes phosphorylated for more than 20 min following pH shock to catalyze MT endwise disassembly to provide tubulins for flagellar regeneration (*Wang et al., 2013*). Thus, the heightened enzyme-mediated MT depolymerization maybe occurs primarily after HA is washed away, and cannot be reported by EB1-NG. Similarly, osmotic or salt stresses activate an atypical tubulin-kinase and phospholipase D, triggering disassembly or reorganization of plant MT system (*Fujita et al., 2013*; *Dhonukshe et al., 2003*). For yeast, sorbitol hypertonicity induces frozen MTs (*Robertson and Hagan, 2008*) and the recovery in 38 min involves a stress-induced MAP kinase. Similar paradigms may be responsible for the resumption of MT dynamic in HA-bathed algae that takes ~55 min (*Figure 4f*).

The electrolyte sensitivity of algal MT system is contrary to the perceived stable MT system in interphase mammalian cells (*Lieuvin et al., 1994*) that have 140 mM $[Na^+]_{ex}$, and 140 mM $[K^+]_{in}$ (reviewed by *Pohl et al., 2013*). Consistent with this, we cannot elicit any obvious changes of EB1-EGFP patterns in mammalian epithelial cells by compression or illumination (*Matov et al., 2010*). Likewise, both Na⁺ and K⁺ promote tubulin polymerization, with 160 mM Na⁺ as the optimal condition (*Olmsted and Borisy, 1975*). Different cation sensitivities could be due to sequence divergence. The other possible cause is the presence of accessory proteins, such as MAPs that obscure the cation sensitivity of mammalian MTs (*Olmsted and Borisy, 1975*; *Wolff et al., 1996*). Alternatively, signaling pathways or the capacity to maintain electrolyte homeostasis could differ. Thus, while fundamental features of the MT system - likely dynamic instability and EB1 plus end tracking - are universal, electrolyte sensitivity and responses could diverge.

## Implications of electrolyte sensitivities of algal MT system

The striking changes in EB1 patterns elicited by H⁺ and Na⁺ at different concentrations may explain intriguing phenomena occurring at disparate scales ranging from cellular compartments to ecosystems. For example, the pH gradient at the pollen tube may account for the establishment of the MT zone lagging the F-actin, Ca²⁺ and acid zone at the very tip (*Gibbon and Kropf, 1994*; reviewed by *Hepler, 2016*). For sea urchins, a basic shift directs MT-supported processes following fertilization. Depressing pH triggers MT disassembly and inhibits division of fertilized zygotes (*Schatten et al., 1985*). This and an increase of 0.3–0.5 pH unit in mitosis inspired the pH clock hypothesis for cell cycle control (*Gagliardi and Shain, 2013*). In line with this, EB1 preferentially binds to MT plus ends in arrested mitotic phase extract of *Xenopus* oocytes, but uniformly decorates MTs in interphase extract (*Tirnauer et al., 2002*), perhaps with a lower pH, analogous to HA-induced bird cage pattern (*Figure 3a*). Given the central roles of EB1 and dynamic MTs for mitosis and the swiftness of pH-induced changes in MTs and EB1 patterns, tuning pH may indeed control cell cycle for certain organisms.

Similarly, absence of most EB1 signals correlates with blocked cell division at 200 mM NaCl (*Figure 7*). Consistent with this, proliferation of commonly used *Chlamydomonas* strains is mostly blocked in the culture media containing NaCl of 200 mM or a higher concentraion (*Perrineau et al., 2014*; *Takouridis et al., 2015*; *Khona et al., 2016*) in the efforts of industrial scale cultures of microalgae with media of high salinity. Adaptation to such salinity takes months and involves mutation, sextual reproduction and evolution. The MT system is likely one of the limiting factors. This study also shows that proliferation is also partially inhibited at 55 and100 mM at least within 24 hr. Although division events are fewer and delayed at 100 mM compared to that at 55 mM, comet lengths and speeds were similar at these two conditions. Perhaps cell division at ~100 mM NaCl is inhibited by additional reactions that may or may not relate to the MT system. Changes occurring below 200 mM may be adapted faster, perhaps simply by metabolic changes (*Husic and Tolbert, 1986*). Intracellular acidification followed by immersions in buffers with merely 5 mM $Na^+$ (*Figure 6a*) may mimic changes in natural environments.

Acidity and salinity are key factors affecting the biota. They vary drastically, often seasonally, among freshwater ecosystems, influenced by a combination of climate changes, biogeochemistry, episodic storm surges, snow melts and anthropological activities (reviewed by *Sullivan, 2000*; *Herbert et al., 2015*). Notably, pH following a downpour could decrease by one pH unit and to ~4.5 for typical watersheds (*Feeley et al., 2013*). Dissolved organic carbons, such as permeant HA, are a major contributor to local acidification, reaching up to mM levels (*Monteith et al., 2007*; *Porcal et al., 2009*; *Zuo et al., 2012*). In addition, acidification caused by acid rains from air pollution primarily in the past (*Sullivan, 2000*) and by the projected increase in atmospheric $CO_2$ in the future (*Raven and Caldeira, 2005*) will also shift the equilibrium toward permeant neutral acids and favor intracellular acidification.

Salinity-dependent effects on *Chlamydomonas* (*Figure 7a*) typify the inverse relationship of salinity and species diversity; and the exceptional vulnerability to salinity of microalgae and zooplanktons that are at the bottom of the food chain (reviewed by *Herbert et al., 2015*). The sub- or hypo-saline level of 55 mM $Na^+$ that perturbs alga's MT system and cell division is readily exceeded by $Na^+$ concentration in numerous inland and coastal wetlands throughout the world from intrusion of seawater (~469 mM $Na^+$), rising of saline ground water and mining effluent discharges (*Herbert et al., 2015*). Similarly, deicing road salt for traffic safety is transforming pristine water bodies in North America, especially urban riverways (*Corsi et al., 2010*; *Dugan et al., 2017*; *Jones et al., 2017*). For example, $Cl^-$, a standard salinity indicator, exceeded 24 mM (860 mg/L), the acute criteria for aquatic life established by United States Environmental Protection Agency, in more than 55% of tested sites at the Milwaukee river basin, a major estuary of the Great Lakes, and was as high as 315 mM (*Corsi et al., 2010*). It is of interest to test how current criteria affect algal cell biology acutely and chronically, and whether this could forecast the feasibility of the ultimate goal in restoring natural reproduction of aquatic animals.

While current concerns about rising atmospheric $CO_2$ largely center on extracellular acidification and hindered shell formation (*Hoegh-Guldberg et al., 2007*; *Waldbusser et al., 2013*; *Fitzer et al., 2016*), this study suggests that the concerns should be extended to intracellular acidification, rising salinity and the MT system. Seemingly harmless rises in salinity in conjunction with chronic mild acidification or a temporary acid surge may poise to threaten the vulnerable unicellular organisms via a perturbed MT system, reshape aqueous landscapes and drive evolution.

Electrolyte-induced rapid changes in the MT system may be applicable to land plants and even particular mammalian cells. Like *Chlamydomonas*, land plant model systems subjected to salt stress, osmostress, or biotic stresses exhibit growth inhibition, MT bundling and heightened sensitivity to microtubule-stabilizing Taxol (e.g. *Dhonukshe et al., 2003*; *Shoji et al., 2006*; *Wang et al., 2007*; *Zhang et al., 2012*; *Fujita et al., 2013*; reviewed by *Hardam, 2013*; *Hashimoto, 2015*; *Oda, 2015*). The similarity comports with their common cortical MTs and a great homology of the non-flagellar proteins in the MT system (*Dymek et al., 2006*; *Pedersen et al., 2003*; *Merchant et al., 2007*; reviewe by *Gardiner, 2013*). MT changes induced by various abiotic stresses, including salt, have prompted an interesting proposition that the MT system is an abiotic sensor of plant cells (*Haswell and Verslues, 2015*; *Wang et al., 2011a*, *2011b*). However, it remains contentious whether the changes of MTs are instead the consequence of salt stress signaling. The EB1-reported scaled responses that seem proportional to $[Na^+]_{in}$ and the speed of manifestations (*Figures 4d* and *5*) in fresh water algae strengthen the possibility that plant MT system is an upstream player in the

salinity signaling pathways, if not the very sensor. In line with this, channel-linked MTs are integral to osmolarity signaling transduction in mammalian osmosensory neurons (*Prager-Khoutorsky et al., 2014*). Using the experimental strategies developed in this study, it is possible to investigate quantitatively the diverged mechanisms of eukaryotic MT systems in sensing and responding to salt stresses.

## Materials and methods

### *Chlamydomonas* strains and culture

The wild type strain CC-124 and a β-tubulin mutant *tub2-1* (formerly *col$^R$4*) (*Bolduc et al., 1988*; *Lee, 1990*; *Schibler and Huang, 1991*) were from *Chlamydomonas* Resource Center (http://www.chlamycollection.org/). They were converted into EB1-NG transgenic strains as described (*Harris et al., 2016*). Cells were cultured in 300 ml pH7.0 standard Tris Aacetate Phosphate (TAP) liquid media with aeration at 25°C over a 14/10 light/dark cycle until reaching logarithmic phase of growth (5–10 $\times$ 10$^6$ cells/ml) (*Sivadas et al., 2012*). All experiments were completed at least 2 hr before the onset of the dark period.

### Solutions

Glacial acetic acid was diluted with ddw to various concentrations ranging from 5 to 1000 mM. The 10 mM HCl solution was titrated to pH3 with 1 M NaOH. The 1000 mM HA was added to the TAP medium to make 20 mM pH4.5 HA/TAP. For 5 mM Na$^+$/HEPES and K$^+$/HEPES, pH of 10 mM HEPES was adjusted to 7.4 with NaOH or KOH respectively. To make 21 mM Na$^+$/EGTA and K$^+$/EGTA solutions, 10 mM EGTA was titrated to pH8 with NaOH or KOH. The 5 mM NaCl and KCl solutions were made by dissolving the respective salt in ddw. NaCl was added into 5 mM pH7.4 Na+/HEPES buffer or TAP liquid media to make 55 to 200 mM Na$^+$/HEPES or Na$^+$/TAP. The solutions of 30 and 75 mM Ca$^{2+}$/HEPES were made by dissolving CaCl$_2$ in 5 mM pH7.4 Na$^+$/HEPES.

### Live cell imaging and treatments

EB1-NG in live *Chlamydomonas* cells was imaged with Nikon Eclipse widefield microscope equipped with a short-arc mercury lamp, an FITC-HYQ optical filter set, a CoolSNAP-ES CCD camera and MetaMorph software. Each image was captured as a 16-bit grayscale file with 1 s exposure. Streaming videos were recorded for 100 frames at a rate of 1 frame/sec.

Typically, cells were resuspended in solutions for 5 min unless indicated otherwise. An aliquot of 5 µl cell suspension was placed on a slide and then covered by a 18 $\times$ 18 mm$^2$ cover slip. The edges were sealed with nail polish before imaging. For compression experiments, a 3 µl aliquot of cell suspension was placed on a glass slide and then covered with a 22 $\times$ 22 mm$^2$ cover slip. Cells became gradually compressed by the coverslip as evident by flattened cell body.

For pH pulse in a perfusion chamber, an aliquot of 10 µl cells in the TAP medium was placed on a cover slip pre-coated with 5 µl 0.001% poly-L-lysine. The cover slip was then inverted to assemble a perfusion chamber as shown in *Figure 2a*. The chamber was flushed with 200 µl 20 mM pH4.5 HA/TAP. Subsequently, HA/TAP was replaced by a flush of 200 µl TAP. The entire process was recorded in two consecutive live-stream clips. For this long recording duration, excitation light intensity was reduced to 25% with a neutral density filter. For HA pulse in a diffusion chamber, 40 µl cells in 5 mM pH7.4 Na$^+$/HEPES was placed at one side of a diffusion chamber underneath a 40X objective lens (*Figure 2a*). A live-streaming video was recorded following the injection of 20 µl 100 mM pH2.8 HA through the Vaseline wall to the opposite side of the chamber. For HA bath, a cell pellet from 50 µl liquid culture was resuspended in 50 µl 10 mM pH3 HA. An aliquot of 10 µl cell mixture was placed on a cover slip. The cell-loaded cover slip was inverted to create a perfusion chamber. After a total 5 min exposure to HA, HA was flushed away with an aliquot of 200 µl-indicated fluid and then a video was recorded. To test MT cold lability after recovery from HA bath, a perfusion chamber with treated cells was chilled by ice for 3 min. A video was taken immediately afterwards, ~20 s after the chamber was removed from ice. For assessing Na$^+$ effects on cell division, right before the onset of the dark cycle at 8 PM, cells pelleted into a 15 ml conical tubes from 30 ml liquid cultures were resuspended with 8 ml TAP media supplemented with NaCl of the concentration as indicated. Triplicates were prepared for each concentration. The tubes were wrapped in aluminum foil and placed

horizontally in the dark room. Under a red safelight, a 500 µl aliquot taken from each tube every two hours until 8:00 AM was fixed with 500 µl 4% glutaraldehyde. A final sample was fixed at 5 PM. A 10 µl aliquot of fixed cells was observed under a 40X objective in an Olympus compound microscope. Numbers of single cells and sporangia were counted in randomly selected fields. More than 500 cell particles were counted for each tube and each time point. The entire experiment was repeated for live fluorescent imaging. A 10 µl aliquot from each tube at each time point were recorded with live streaming as described. Each treatment was repeated at least twice in each experiment. Individual experiments were repeated independently three times at least.

### Image analysis

To measure EB1 comet speed, a 40 s substack containing side views of cells were first made by the open source image process software, ImageJ (https://imagej.nih.gov/ij/index.html); and individual comets were analyzed with a Matlab-based particle tracking software, plusTipTracker (*Applegate et al., 2011*). In each cell that maintained completely quiescent for the tracking period, all tractable comets which transverse at least one third of the cell length were analyzed. The numbers of qualified cells and comets from numerous recordings were indicated. To generate line scans of EB1 intensity at microtubule plus ends, a line tool in ImageJ was used to measure gray values along the length of comets. Relative fluorescence intensity was normalized after calculation by subtracting a background gray value measured next to the comet with the line tool. Histograms were generated with the Microsoft program, Excel. Kymographs were generated with an ImageJ plug-in multiple kymograph (https://www.embl.de/eamnet/html/body_kymograph.html).

### Statistical analysis

All data are given as mean ±SEM (standard error of the mean) and analyzed with Sigmaplot 13.0 (Systat Software, Inc., San Jose, CA). Sample sizes for comet speed measurement were limited by the fact that few cells are entirely quiescent, which is necessary for digital tracking.

# Acknowledgement

This work is supported by Marquette University Startup for P Yang; and by a grant from the National Institutes of Health (R01GM110413 to KL). The content is solely the responsibility of the authors and does not necessarily represent the official views of the National Institutes of Health. .

# Additional information

## Funding

| Funder | Grant reference number | Author |
|---|---|---|
| National Institutes of Health | GM110413 | Karl F Lechtreck |
| National Institutes of Health | R01GM110413 | Karl F Lechtreck |
| Marquette University | Start up | Pinfen Yang |
| Marquette University | RCM Growth Incentive Fund | Pinfen Yang |

The funders had no role in study design, data collection and interpretation, or the decision to submit the work for publication.

## Author contributions

Yi Liu, Conceptualization, Data curation, Software, Formal analysis, Supervision, Validation, Investigation, Visualization, Writing—original draft, Writing—review and editing; Mike Visetsouk, Investigation, Methodology; Michelle Mynlieff, Conceptualization, Methodology, Writing—review and editing; Hongmin Qin, Investigation, Methodology, Writing—review and editing; Karl F Lechtreck, Conceptualization, Resources, Writing—review and editing; Pinfen Yang, Conceptualization, Resources, Supervision, Funding acquisition, Investigation, Methodology, Writing—original draft, Project administration, Writing—review and editing

## Author ORCIDs

Yi Liu, http://orcid.org/0000-0002-9529-208X
Karl F Lechtreck, http://orcid.org/0000-0002-6219-6470
Pinfen Yang, http://orcid.org/0000-0002-3773-0053

## Decision letter and Author response

Decision letter https://doi.org/10.7554/eLife.26002.018
Author response https://doi.org/10.7554/eLife.26002.019

## Additional files

**Supplementary files**
• Transparent reporting form
DOI: https://doi.org/10.7554/eLife.26002.017

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
