## [Decision Letter]

Thank you for submitting your article "H^+^- and Na^+^- elicited swift changes of the microtubule system in the biflagellated green alga *Chlamydomonas*" for consideration by *eLife*. Your article has been reviewed by three peer reviewers, one of whom is a member of our Board of Reviewing Editors, and the evaluation has been overseen by Ian Baldwin as the Senior Editor. The following individual involved in review of your submission has agreed to reveal his identity: Junmin Pan (Reviewer #2).

The reviewers have discussed the reviews with one another and the Reviewing Editor has drafted this decision to help you prepare a revised submission.

Summary:

This paper demonstrates, for the first time, in vivo visualization of cytoplasmic microtubule dynamics in *Chlamydomonas*. Using a EB1-neon-green fluorescent probe, the authors measure how microtubule dynamics change as a function of changing external ionic conditions. The data are striking and do indeed support the primary claim of the manuscript, that cytoplasmic microtubules in *Chlamydomonas* show dynamics that is sensitive to specific chemical (ionic and pH) conditions.

Essential revisions:

1) Although the authors describe interesting phenomena, the biological significance is not clear. The authors need to establish the biological significance, preferentially through additional experiments. For example, do the extracellular ionic and pH changes alter cell behavior such as cell growth, swimming, phototaxis, etc. or lead to ultrastructural changes? While the relationship between any such alterations and the observed changes in microtubule dynamics would only be correlative, it could nevertheless suggest a biological function.

2) Also with respect to the biological relevance, could the authors provide some estimates of how much things like sodium, calcium, and pH change in normal freshwater ponds, for example after a heavy period of rain, or in late summer when the water level has dropped.

3) How do the results relate to reports from the Pan lab (Wang et al., 2013) showing shortening of cytoplasmic microtubules during flagellar regeneration after pH shock. In the Discussion the MT shortening reported by Wang et al. is described as an ionic response, but that wasn't how the authors of the cited study described their results, who instead interpreted the result as competition between flagella and cytoplasm for tubulin. Perhaps what would be useful is if the authors could give some indication of what is happening with flagella in their experiments, i.e. in their complicated multi-wash experiments, when do flagella start regenerating if at all?

4) The logical flow is not clear. The paper is hard to follow as the authors go back and forth between different treatments (HA and Ca^2+^ treatment), strains (wt, tub2) and phenomena (comets, bird cage pattern, thick microtubules). The reviewers suggest an extensive rewriting, with the inclusion of a table to summarize the main results.

Title and Abstract

I suggest a slight change in the title: "H^+^- and Na^+^- elicited rapid changes in the microtubule cytoskeleton in the biflagellated green alga *Chlamydomonas*"

The Abstract needs to be rewritten. Mentioning interphase animal cells seems irrelevant. What does "mercurial" mean and what relevance does it have to the current results. I would not use the term "fluctuation" but rather "change" or "alteration". The last sentence needs to be deleted: the author do not measure changes in the microtubule cytoskeleton in animal cells in response to changes in ionic conditions; they do not directly show that pH drives cellular processes (extracellular pH is changed, but the change in intracellular pH is not measured and it is not established that intracellular pH change that causes the change); the paper does not study "many species" (only one), so the last point is unsupported.

[Editors' note: further revisions were requested prior to acceptance, as described below.]

Thank you for resubmitting your work entitled "H^+^- and Na+- elicited rapid changes of the microtubule cytoskeleton in the biflagellated green alga *Chlamydomonas*" for further consideration at *eLife*. Your revised article has been favorably evaluated by Ian Baldwin (Senior editor), a Reviewing editor, and two reviewers.

The manuscript has been improved but there are some remaining issues that need to be addressed before acceptance, as outlined below:

1) Regarding salinity information – in their rebuttal letter, the authors argue that results can vary between bodies of water, but we don't see why that is a reason not to report some results, for at least some bodies of water. The main question is whether the salinity ranges they use in their study are ever reached in natural conditions; the fact that studies are mentioned but none are cited to support their conditions as being relevant is worrying. Contrast this with pH where they make a very compelling case that they are in a biologically relevant pH range. Please make a stronger effort to address this issue.

2) For the minor point, we still do not agree with the description of "two pairs of basal body": there are two basal bodies, each basal body is associated with a procentriole. Thus, we do not think this description is correct.

---

## [Author Response]

Essential revisions:1) Although the authors describe interesting phenomena, the biological significance is not clear. The authors need to establish the biological significance, preferentially through additional experiments. For example, do the extracellular ionic and pH changes alter cell behavior such as cell growth, swimming, phototaxis, etc. or lead to ultrastructural changes? While the relationship between any such alterations and the observed changes in microtubule dynamics would only be correlative, it could nevertheless suggest a biological function.

The biological significance of Na^+^ – induced changes in *Chlamydomonas* is provided by the new figure, Figure 7, that demonstrates a partial correlation of a perturbed microtubule system and cell division with increasing Na^+^ concentrations. Discussion of this topic is now consolidated in a separate section “Implications of electrolyte sensitivities of algal microtubule system**”**.

How extracellular ions and pH affect flagellar excision and flagellar motility were shown or described in Figure 2, Figure 4 and Figure 6; and established in the *Chlamydomonas* biology. We refrain from linking the changes in the cell body microtubules and flagellar motility, since motility machinery could operate without the cell body. Furthermore, electrolytes and pH could potentially affect directly the enzyme-powered machinery.

Illumination that is upstream to phototaxis was mentioned in Figure 1. We did not test or discuss phototaxis further, since flagellar motility is required for phototaxis, and light sensing that directs flagellar motility is upstream to the changes of the microtubules. Therefore, effects of electrolytes and pH on phototaxis, if any, cannot be attributed to changes in the cell body microtubules.

2) Also with respect to the biological relevance, could the authors provide some estimates of how much things like sodium, calcium, and pH change in normal freshwater ponds, for example after a heavy period of rain, or in late summer when the water level has dropped.

This suggestion inspired us to search for relevant ecological data. We learned that fresh water streams, ponds and lakes vary drastically, and often seasonally, in ionic compositions, organic acids (dissolved organic carbons) and pH due to storm surges, biogeochemistry, climate changes and anthropological activities, such as air pollution and road salting. Therefore, data from one waterbody cannot be truly representative. Instead, we discuss how our data are quantitatively relevant to the pressing acidification and salinization of aquatic environments. We believe that such comparisons are appropriate, interesting and important to concerned global citizens. We thank the reviewer for this great question.

3) How do the results relate to reports from the Pan lab (Wang et al., 2013) showing shortening of cytoplasmic microtubules during flagellar regeneration after pH shock. In the Discussion the MT shortening reported by Wang et al. is described as an ionic response, but that wasn't how the authors of the cited study described their results, who instead interpreted the result as competition between flagella and cytoplasm for tubulin. Perhaps what would be useful is if the authors could give some indication of what is happening with flagella in their experiments, i.e. in their complicated multi-wash experiments, when do flagella start regenerating if at all?

We thank the reviewer’s insight. This point indeed was not discussed explicitly in the manuscript. The changes address all pertinent questions:

“Some electrolyte-elicited changes could be evoked by signaling pathways in *Chlamydomonas* as in other organisms (Wang et al., 2013; Perrineau et al., 2014; Khona et al., 2016), which perhaps occur later and slower; and last longer than the direct effects from electrolytes. For example, kinesin-13 becomes phosphorylated for more than 20 min following pH shock to catalyze MT endwise disassembly to provide tubulins for flagellar regeneration (Wang et al., 2013). Thus, the heightened enzyme-mediated MT depolymerization maybe occurs primarily after HA is washed away, and cannot be reported by EB1-NG. […]”

A sentence is added in the Results section:

“Flagellar regeneration was not assessed because both chambers were not suitable for the long regeneration process.”

4) The logical flow is not clear. The paper is hard to follow as the authors go back and forth between different treatments (HA and Ca^2+^ treatment), strains (wt, tub2) and phenomena (comets, bird cage pattern, thick microtubules). The reviewers suggest an extensive rewriting, with the inclusion of a table to summarize the main results.

The original version describes first wild type cells and then *tub2* mutant cells. Since it is hard to follow the story line, we rearrange the flow, describing how both strains responding to each treatment sequentially. As such, *tub2* data in Figure 3 are now split into two figures. Table 1 is added for easy comparisons of changes elicited by major treatments. The text, legends and Materials and methods have been changed to reflect the new order, new figures and the additional table.

Title and AbstractI suggest a slight change in the title: "H^+^- and Na^+^- elicited rapid changes in the microtubule cytoskeleton in the biflagellated green alga Chlamydomonas"

Changed.

The Abstract needs to be rewritten. Mentioning interphase animal cells seems irrelevant. What does "mercurial" mean and what relevance does it have to the current results. I would not use the term "fluctuation" but rather "change" or "alteration". The last sentence needs to be deleted: the author do not measure changes in the microtubule cytoskeleton in animal cells in response to changes in ionic conditions; they do not directly show that pH drives cellular processes (extracellular pH is changed, but the change in intracellular pH is not measured and it is not established that intracellular pH change that causes the change); the paper does not study "many species" (only one), so the last point is unsupported.

Abstract has been revised substantially. Specifically, “Interphase animal cells” deleted. “[…] mercurial[…]” and “[…]fluctuation of[…]” are rephrased. The last sentence is now more measured.

“Although microtubules are known for dynamic instability, the dynamicity is considered to be tightly controlled to support a variety of cellular processes. […]These results from this model organism with characteristics of animal and plant cells provide novel explanations regarding how pH may drive cellular processes; how plants may respond to, and perhaps sense stresses; and how organisms with a similar sensitive cytoskeleton may be susceptible to environmental changes.”

[Editors' note: further revisions were requested prior to acceptance, as described below.]

The manuscript has been improved but there are some remaining issues that need to be addressed before acceptance, as outlined below:1) Regarding salinity information – in their rebuttal letter, the authors argue that results can vary between bodies of water, but we don't see why that is a reason not to report some results, for at least some bodies of water. The main question is whether the salinity ranges they use in their study are ever reached in natural conditions; the fact that studies are mentioned but none are cited to support their conditions as being relevant is worrying. Contrast this with pH where they make a very compelling case that they are in a biologically relevant pH range. Please make a stronger effort to address this issue.

The original paragraph about the impact of pH and salinity is split into two. The new paragraph solely devoted to deliberate salinity quantitatively is:

“Salinity-dependent effects on *Chlamydomonas* (Figure 7) typify the inverse relationship of salinity and species diversity; and the exceptional vulnerability to salinity of microalgae and zooplanktons that are at the bottom of the food chain (reviewed by Herbert et al., 2015). […] It is of interest to test how current criteria affect algal cell biology acutely and chronically, and whether this could forecast the feasibility of the ultimate goal in restoring natural reproduction of aquatic animals.”

A new paper is added to the reference list.

2) For the minor point, we still do not agree with the description of "two pairs of basal body": there are two basal bodies, each basal body is associated with a procentriole. Thus, we do not think this description is correct.

The simplified original statement was revised into:

“Flagellated *Chlamydomonas* cells in interphase contain two mature basal bodies (BBs) and two probasal bodies (Figure 1, top panel) that are templated on mature BBs. Each of mature BBs that are derived from centrioles following cell division nucleates the assembly of the axoneme, a MT-based scaffold that drives the rhythmic beating of the flagellum (Dutcher and O’Toole et al., 2016).”

The expertly written review is added to the reference list.